# How hibernation in frogs drives brain and reproductive evolution in opposite directions

Wenbo Liao[1,2,3]*, Ying Jiang[1,2,3], Long Jin[1,2,3], Stefan Lüpold[4]*

[1]Key Laboratory of Southwest China Wildlife Resources Conservation (Ministry of Education), China West Normal University, Sichuan, China; [2]Key Laboratory of Artificial Propagation and Utilization in Anurans of Nanchong City, China West Normal University, Nanchong, China; [3]Institute of Eco-Adaptation in Amphibians and Reptiles, China West Normal University, Nanchong, China; [4]Department of Evolutionary Biology and Environmental Studies, University of Zurich, Zurich, Switzerland

**Abstract** Environmental seasonality can promote the evolution of larger brains through cognitive and behavioral flexibility but can also hamper it when temporary food shortage is buffered by stored energy. Multiple hypotheses linking brain evolution with resource acquisition and allocation have been proposed for warm-blooded organisms, but it remains unclear how these extend to cold-blooded taxa whose metabolism is tightly linked to ambient temperature. Here, we integrated these hypotheses across frogs and toads in the context of varying brumation (hibernation) durations and their environmental correlates. We showed that protracted brumation covaried negatively with brain size but positively with reproductive investment, likely in response to brumation-dependent changes in the socio-ecological context and associated selection on different tissues. Our results provide novel insights into resource allocation strategies and possible constraints in trait diversification, which may have important implications for the adaptability of species under sustained environmental change.

*For correspondence: liaobo_0_0@126.com (WL); stefan.luepold@ieu.uzh.ch (SL)

Competing interest: The authors declare that no competing interests exist.

## eLife assessment

In this **important** paper, the authors report a link between brumation (or "hibernation") and tissue size in frogs, summarizing **convincing** evidence that extended brumation is associated with smaller brain size and increased investment in reproduction-related tissues. The research is of broad interest to ecologists, evolutionary biologists, and those interested in global change biology, as the dataset involves significant field work and advanced statistical analyses for insights into how expensive tissues in these ectothermic animals respond to environmental seasonality.

## Introduction

Seasonal food scarcity challenges animal energy budgets. A positive or less negative energy balance across seasons can be achieved by a more constant net energy intake than predicted solely by food abundance (*Sol, 2009*), or by investing less in costly organs (*Heldstab et al., 2018*). A link between both strategies is the brain: A relatively large brain can improve cognitive ability and behavioral flexibility (*Benson-Amram et al., 2016*; *Lefebvre et al., 2004*; *Reader and Laland, 2002*; *Sol et al., 2005*), enabling animals to effectively locate diverse and dispersed food sources to buffer environmental fluctuations in seasonal habitats (cognitive buffer hypothesis) (*Allman et al., 1993*; *Sol, 2009*;

van Woerden et al., 2012). However, brain tissue has high metabolic costs (*Aiello and Wheeler, 1995*; *Lukas and Campbell, 2000*; *Mink et al., 1981*) that may not be temporarily reducible (*Mink et al., 1981*), constraining brain size evolution under periodic food scarcity (expensive brain hypothesis) (*Isler and van Schaik, 2009*).

Food scarcity can also affect physiological responses. For example, a longer digestive tract may permit more efficient resource uptake during a short active period and be favored by selection in species with prolonged hibernation (*Sibly, 1981*). The evolution of the digestive tract could thus parallel that of the brain. In contrast, there could be an evolutionary trade-off between the two organs ('expensive tissue hypothesis'; *Aiello and Wheeler, 1995*; also see *Isler and van Schaik, 2006* for a more general 'energy trade-off hypothesis'). Furthermore, physiological buffering often involves a seasonal reduction in metabolic rate or activity (e.g., hibernation), with energy drawn from stored fat reserves (*Heldstab et al., 2016*). Buffering lean periods by fat stores and reduced activity contrasts with cognitive abilities to cope with food scarcity, suggesting differential investment in adipose and brain tissue between species in highly seasonal environments ('fat–brain trade-off hypothesis'; *Heldstab et al., 2016*; *Navarrete et al., 2011*). Finally, allocating fat stores to sustained brain function throughout prolonged hibernation might compete with investments in other tissues (*Isler, 2011*; *Isler and van Schaik, 2009*).

Species that breed soon after hibernation, such as some mammals (*Place et al., 2002*; *Psenner, 1957*) and many amphibians (*Fei and Ye, 2001*; *Wells, 1977*), face further unique challenges, as their reproductive tissue must regrow before emergence when stored resources are most limited (*Isler, 2011*; *Isler and van Schaik, 2009*). Testes, however, may be subject to intense selection by sperm competition resulting from female multiple mating (*Lüpold et al., 2020*), a widespread phenomenon where males cannot monopolize their mates (*Lüpold et al., 2014*). Indeed, across anurans (frogs and toads) that often breed in small water bodies, males invest relatively more in their testes and less in their forelimbs (used in pre-mating competition) as population density increases (*Buzatto et al., 2015*; *Lüpold et al., 2017*). If a shorter active period leads to more synchronized breeding and thus increases the risk of sperm competition or sperm depletion (*Vahed and Parker, 2012*), selection for relatively larger testes would be stronger where fat stores need to last longer, affecting resource demands and allocation during hibernation. Including reproductive investments and breeding patterns in studies of allocation trade-offs in response to hibernation and environmental seasonality would thus seem critical but remains to be done, particularly in the context of brain evolution.

The opposing selection pressures on brain or gonad size (i.e., cognitive or fitness benefits versus metabolic costs), varying degrees of seasonality and diverse strategies for buffering periodic food scarcity (e.g., cognitive versus physiological) between species render environmental fluctuations an ideal context for studying brain size evolution. The different hypotheses for the coevolution of brain size with other organs were developed separately for different mammalian or avian taxa, the two vertebrate classes with the largest brains relative to body size (*Jerison, 1973*). These hypotheses have yet to be directly tested against one another in a single taxon and ideally in the immediate context of seasonal activity, considering the extent, rather than the mere presence/absence, of hibernation. This last point is important because 'hibernation' can range between brief inactive bouts and long dormancy, with different energetic and life-history constraints. Furthermore, understanding the generality of the patterns reported in mammals or birds needs to be validated in other taxa, ideally with smaller brains and different energy demands. Such a generalization would help contextualize brain evolution in the two largest brained taxa in relation to selection on encephalization and metabolic constraints.

A particularly suitable system is presented in ectothermic (cold-blooded) species whose metabolism and activity are tightly linked to ambient temperature (*Wells, 2007*), possibly leading to stricter physical boundaries to behavioral flexibility than in the endothermic (warm-blooded) mammals or birds. Compared to the divergent groups of mammals, for which most hypotheses on brain evolution have been proposed, anurans are also more homogeneous in body size and shape, diet, and locomotion (*Kardong, 2019*), while still showing strong environmental effects on brain, reproductive, and other investments (*Liao et al., 2022*; *Lüpold et al., 2017*; *Wells, 2007*). Being less influenced by Bauplan and lifestyle, evolutionary trade-offs with brain size could thus be easier to isolate in anurans than in mammals.

Here, we studied across 116 anuran species how varying 'hibernation' periods and their environmental correlates affect male investments in brain, testes, body fat, limb muscles, and the main visceral organs (see *Figure 1*). Unlike mammals, anurans do not actively depress their metabolism but rather stop activity when ambient temperatures drop below the activity range (referred to here as 'brumation' for distinction; *Pinder et al., 1992*; *Wells, 2007*). However, in both taxa, hibernators use fixed energy stores across sequential investments, albeit at a lower metabolic rate (*Staples, 2016*), while non-hibernators can partly replenish resources but need more food even when it is scarce (*Heldstab et al., 2016*). These contrasting strategies may be associated with a varying cost–benefit balance between organs, including the brain.

Although anurans halt general activity during brumation, this does not mean complete brain inactivity. Several species move in their burrows or underwater hibernacula in response to changes in soil temperature or oxygen concentration (*Holenweg and Reyer, 2000*; *Stinner et al., 1994*; *van Gelder et al., 1986*), or when disturbed (*Niu et al., 2022*; *Tattersall and Ultsch, 2008*). Moreover, brumating frogs can renew more brain cells than active frogs, possibly to prevent brain damage (*Cerri et al., 2009*), and this could increase with brain size. Larger brains could also be less tolerant to low oxygen conditions (*Pinder et al., 1992*; *Tattersall and Ultsch, 2008*; *Wells, 2007*) if the findings from other ectotherms extend to anurans (*Sukhum et al., 2016*). Overall, maintaining a relatively larger brain while brumating may entail higher costs that could constrain brain evolution compared with species with short or no seasonal inactivity.

Instead of the typical indirect proxies or scores (*Heldstab et al., 2018*; *Heldstab et al., 2016*; *Luo et al., 2017*; *Navarrete et al., 2011*), we quantified seasonal changes in tissue size and estimated brumation duration as the time that temperatures were continuously below the species-specific activity threshold. These estimates aligned with periods of inactivity recorded during field surveys, and were robust across multiple sensitivity analyses (e.g., considering temperature buffering by underground burrows). We then tested the different hypotheses on brain evolution and further integrated variation in breeding context and reproductive evolution in this resource allocation framework. By examining ectothermic species with diverse energy demands and brain sizes, our findings provide a crucial context for understanding brain evolution and resource allocation.

## Results

### Determinants of brumation duration

For a mean (± standard deviation) of 3.41 ± 0.95 males in each of 116 anuran species, we combined experimentally determined thermal activity thresholds with multi-year temperature fluctuations at their collection sites to estimate the species-specific brumation periods (details on methodology and various sensitivity analyses for validation in *Materials and methods*). These periods averaged between 0.6 ± 0.5 and 250.5 ± 16.7 days across the 5 years examined, with high repeatability within species (*R* = 0.95 [95% confidence interval, CI: 0.93, 0.96]; *Appendix 1—figure 1*; *Appendix 1—table 1*). These brumation periods largely fell into three groups with ≤9 (N = 22 species), 18–27 (N = 3), and ≥47 days of expected brumation (N = 91), respectively. As the ground microclimate may buffer some of the fluctuations in air temperature and frogs can endure short cold spells without dormancy, we conservatively considered those 25 species with ≤27 days below their experimental temperature threshold unlikely to show any sustained brumation.

We employed phylogenetic generalized least-squares (PGLS) models (*Freckleton et al., 2002*; *Ho and Ané, 2014*) to dissect the factors driving brumation duration. Our analyses indicated that the brumation period increased with the latitude and elevation of study sites, along with the extent of annual temperature fluctuations ($r \geq 0.42$, $t_{114} \geq 5.00$, p <.001; *Appendix 1—table 2*). In contrast, brumation duration was inversely correlated with the annual mean temperature and precipitation, and dry season length ($r < -0.29$, $t_{114} < -3.24$, p < 0.002), but not significantly covarying with longitude or annual fluctuations in precipitation ($r <0.09$, $t_{114} < 0.95$, p > 0.34; *Appendix 1—table 2*). Except for the duration of the dry season, these results persisted for those 91 species with a high probability of sustained brumation (see above; *Appendix 1—table 3*). Among these species, those from cooler and more seasonal climates exhibited lower entry and emergence temperatures from their inactive state (*Appendix 1—table 4*), suggesting heightened cold tolerance to maximize their active period.

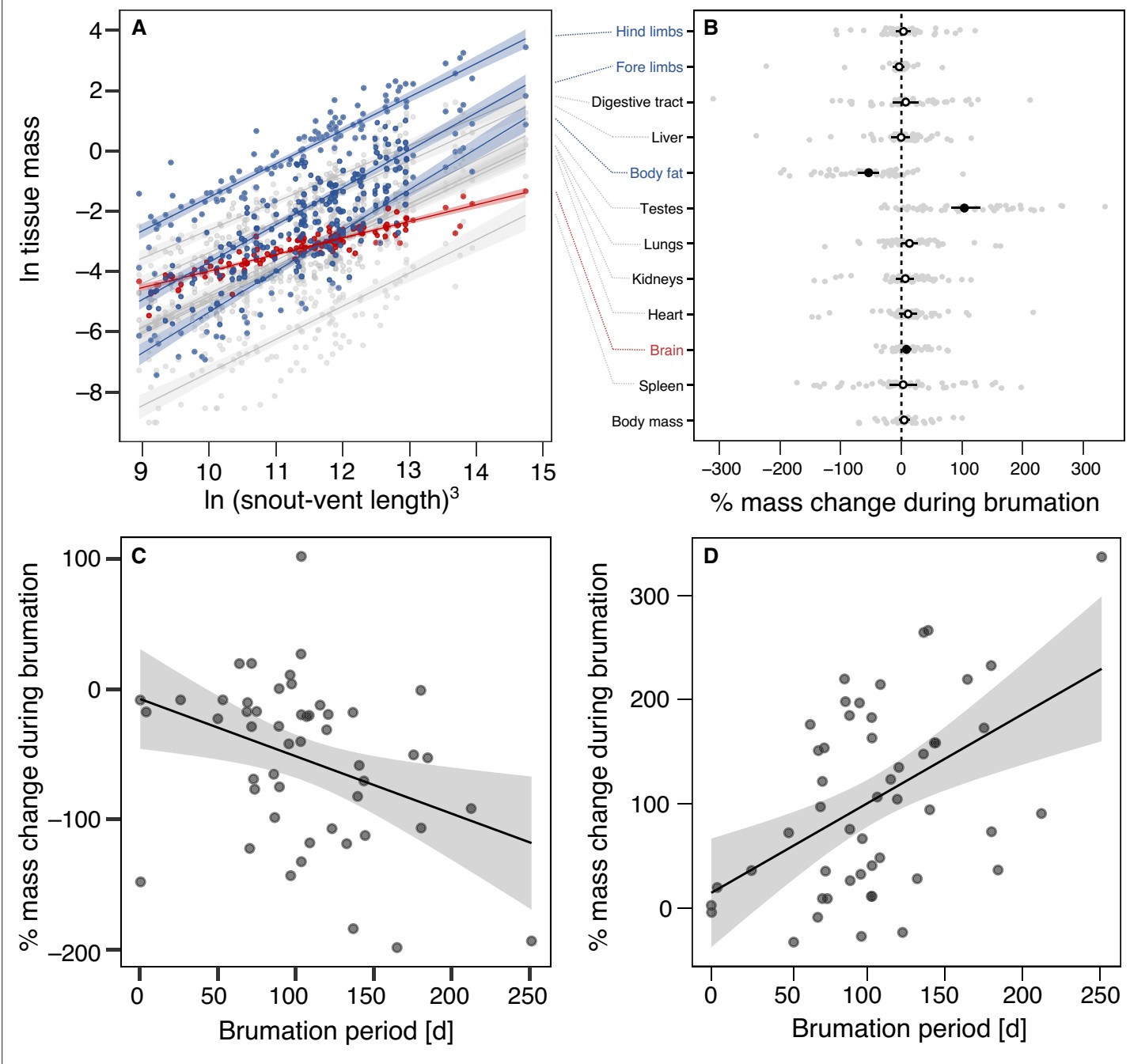

**Figure 1.** Allometric and seasonal variation in the species-specific tissue masses. (**A**) Allometric slopes between the mass of each tissue and cubed snout-vent length (SVL³) so that proportionate scaling follows a slope of 1 on a log–log scale. Each point represents a species-specific mean value in breeding condition (*N* = 16). Relationships deviating from proportionate scaling (based on bootstrapped 95% confidence intervals) are highlighted in blue (steeper than unity) or red (shallower than unity). (**B**) Mean percent change with 95% confidence interval for body mass and each individual tissue of 50 anuran species with data from both shortly before and after brumation (=breeding), based on absolute tissue masses between stages and log-transformed to maintain symmetry and additivity (***Törnqvist et al., 1985***): log(post-brumation/pre-brumation) × 100. The transparent gray dots depict species-specific values. (**C**) Relationship between brumation period and percent mass change in the amount of body fat. (**D**) Relationship between brumation period and percent mass change in testis mass. Each point indicates a species.

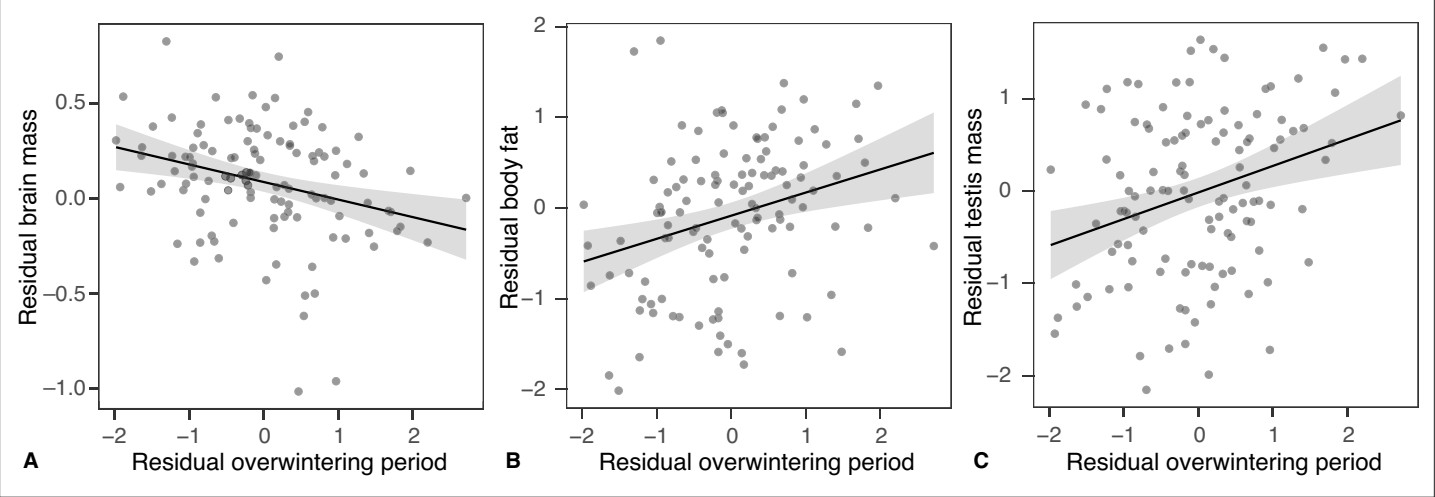

**Figure 2.** Effects of brumation duration on the relative tissue sizes. Relationships between brumation duration and the relative mass of the brain (**A**), body fat (**B**), and testes (**C**) across males of 116 anuran species in breeding (post-brumation) condition. All axes are controlled for the snout-vent length and phylogeny.

## Impact of brumation on tissue investments

To investigate the repercussions of brumation on individual tissue investments, we used the same males from our 116 species. In separate sets of PGLS models, neither snout-vent length (SVL) nor body mass covaried with brumation duration or any other environmental variable ($|r| \leq 0.15$, $|t_{114}| \leq 4.03$, p $\geq 0.11$; *Appendix 1—table 5*), except for a weak, non-statistically significant trend toward reduced body mass at higher elevations ($r = -0.18$, $t_{114} = -1.91$, p = 0.06, phylogenetic scaling parameter $\lambda = 0.94$ [95% CI: 0.85, 1.00]). The different tissues increased with body size, albeit with different allometric slopes. Body fat and limb muscles showed steeper scaling (all $\beta \geq 0$ [1.03, 1.18]), brain size exhibited a shallower slope ($\beta = 0.49$ [0.44, 0.54]; *Appendix 1—table 6*; *Figure 1A*), and the remaining tissues did not deviate from proportionate scaling (i.e., 95% CI including 1.00; *Appendix 1—table 6*; *Figure 1A*). Hence, the evolution of brain size appears to be more constrained than that of other organs when selection favors larger body size.

Given these distinct allocation patterns between brain and other tissues, we next tested whether extended brumation constrains brain size evolution, akin to suggestions for mammals based on presence/absence of hibernation (*Heldstab et al., 2018*). In PGLS models, absolute brain size was independent of brumation duration ($r = -0.10$, $t_{115} = -1.08$, p = 0.28, $\lambda = 0.89$ [0.73, 0.97]), but accounting for SVL as a body size proxy, species with prolonged brumation had relatively smaller brains, quantified during breeding (partial $r$, $r_p = -0.31$, $t_{113} = -3.50$, p < 0.001, $\lambda = 0.35$ [0.00, 0.61]; *Figure 2A*, *Appendix 1—table 7*) or pre-hibernation ($N = 50$ species means based on 2.64 ± 0.94 males each: $r_p = -0.51$, $t_{47} = -4.03$, p < 0.001, $\lambda = 0.00$ [0.00, 0.56]; *Appendix 1—table 7*). This trend was distinct from body size changes in response to brumation, as SVL remained independent of brumation duration (*Appendix 1—table 5*). On average, brumating species tended to possess relatively smaller brains compared to those less likely to experience prolonged brumation (*Appendix 1—table 8*), aligning with mammalian studies utilizing hibernation presence/absence (*Heldstab et al., 2018*). This trend extended to those 91 species categorized as brumating for some period (*Appendix 1—table 9*), reinforcing the link between brumation and brain evolution beyond coarse binary classification. This pattern further persisted when recalculating brumation periods using conservative thresholds of 2 or 4°C below their experimentally derived thresholds, simulating shelter buffering (details and validation in *Materials and methods*; *Appendix 1—table 10*).

Species displaying extended brumation periods further exhibited relatively more body fat ($r_p \geq 0.25$, $t_{113} \geq 2.72$, p $\leq 0.008$; *Figure 2B*) and, particularly during breeding, had relatively larger testes ($r_p = 0.36$, $t_{113} = 4 .06$, p < 0.001, $\lambda = 0.77$ [0.40, 0.90]; *Figure 2C*), along with relatively smaller hindleg muscles ($r_p = -0.22$, $t_{113} = -2.37$, p = 0.02, $\lambda = 0.22$ [0.00, 0.51]; *Appendix 1—table 7*). These patterns generally held true when using the presence/absence of brumation (*Appendix 1—table 8*), buffered temperature fluctuations (*Appendix 1—table 10*), or excluding the 25 species unlikely to brumate

(except for the non-significant effect on body fat; *Appendix 1—table 9*). Other tissue sizes remained independent of brumation (*Appendix 1—Tables 7–10*) and showed not significant changes between sampling periods (*Figure 2*).

Comparing pre- and post-brumation males, averaged across all species, we found a roughly 50% reduction in fat tissue and a 100% increase in testis size, indicating resource depletion and testicular regrowth during brumation, respectively (*Figure 1B*). Only brain size deviated from zero among the remaining tissues, but this change was minimal compared to the body fat and testes, and within the range of many unchanged tissues. Thus, the biological significance of this putative increase in brain size during brumation remains uncertain, possibly reflecting general differences between sampled individuals. We will thus refrain from further interpretation. Among the two tissues with considerable change, the extent of fat depletion increased significantly with the brumation period ($r = -0.36$, $t_{48} = -2.68$, p = 0.01, $\lambda = 0.00$ [0.00, 0.55]), as did that of testis regrowth ($r = 0.39$, $t_{48} = 2.89$, p = 0.006, $\lambda = 0.82$ [0.16, 0.98]; *Figure 1C*).

To examine whether the increase in relative testis size with prolonged brumation might be mediated by a shorter, more synchronized mating season (*Wells, 2007*), we tested for links between brumation duration and different breeding parameters. Prolonged brumation notably shortened the breeding season ($r = -0.57$, $t_{41} = -4.47$, p < 0.001, $\lambda = 0.00$ [0.00, 0.38]; *Appendix 1—figure 2A*), exerting a stronger effect than climatic variables (*Appendix 1—table 11*), particularly when considered together (*Appendix 1—table 12*). Hence, the effect of these climatic variables may be mediated by brumation. Furthermore, a shorter breeding season increased the probability of dense breeding aggregations (phylogenetic logistic regression: $N = 42$, $z = -3.03$, p = 0.002, $\alpha = 0.02$; *Appendix 1—figure 2B*). That brumation might mediate climatic impacts on breeding aggregations through the duration of the breeding season was also supported by a phylogenetic confirmatory path analysis (*Gonzalez-Voyer and Hardenberg, 2014*; *von Hardenberg and Gonzalez-Voyer, 2013*; *Appendix 1—figure 3*; *Appendix 1—table 13*). Finally, when combining these data with published data on the density of breeding populations (*Lüpold et al., 2017*; $N = 8$ species overlapping), a trend emerged toward higher mean population densities in species with shorter breeding seasons ($r = -0.69$, $t_6 = -2.37$, p = 0.06, $\lambda = 0.00$ [0.00, 1.00]), albeit with a small sample size (*Appendix 1—figure 2C*).

To explore possible causal links between breeding parameters and relative testis size, we employed directional tests of trait evolution (*Pagel, 1994*; *Revell, 2012*), assessing whether changes in two binary traits are unilaterally or mutually dependent, or independent (*Pagel, 1994*). To this end, we converted our continuous to binary variables (see *Materials and methods*). Relating small/large testes to short/long breeding seasons, the best-supported scenario based on the Akaike information criterion (AIC) was independent evolution. Yet, the model with changes in relative testis size dependent on those in the breeding season found similar support ($\Delta AIC = 0.83$, $w_{AIC} = 0.33$ compared to independent model with $w_{AIC} = 0.50$; *Appendix 1—figure 4A*), differing from the remaining two models with clearly higher AIC scores ($\Delta AIC \geq 3.40$, $w_{AIC} \leq 0.09$). In another directional test, between relative testis size and aggregation formation, the independent model was the best upported ($w_{AIC} = 0.66$), followed by the scenario of increased relative testis size in response to aggregation formation (*Appendix 1—figure 4*), albeit above a $\Delta AIC$ cut-off of 2 ($\Delta AIC = 2.38$, $w_{AIC} = 0.20$), with the remaining models being clearly less supported ($\Delta AIC \geq 3.64$, $w_{AIC} \leq 0.11$). Hence, it is at least possible that breeding conditions mediate the positive relationship between brumation duration and relative testis size in our relatively small sample of species, while a response of breeding conditions to variation in relative testis size is clearly rejected by our analyses.

## Covariation between tissues

As all tissues rely on the same finite resources, their responses to brumation are likely interconnected. Pairwise partial correlations, controlling for SVL and phylogeny, revealed positive covariation or non-significant associations among all tissue masses (*Appendix 1—figure 5*). As such, our data do not support the expensive tissue (*Aiello and Wheeler, 1995*) or the more general energy trade-off hypotheses (*Isler and van Schaik, 2006*), which predict brain size trade-offs with the digestive tract or other costly organs, respectively. However, since brain size differed from fat, hindlimb muscles and testes in allometric relationships and responses to brumation, pairwise correlations may not capture more complex allocation. To examine the relative investments in these four most informative tissues simultaneously, we represented total body mass as proportions of these tissues and the remaining

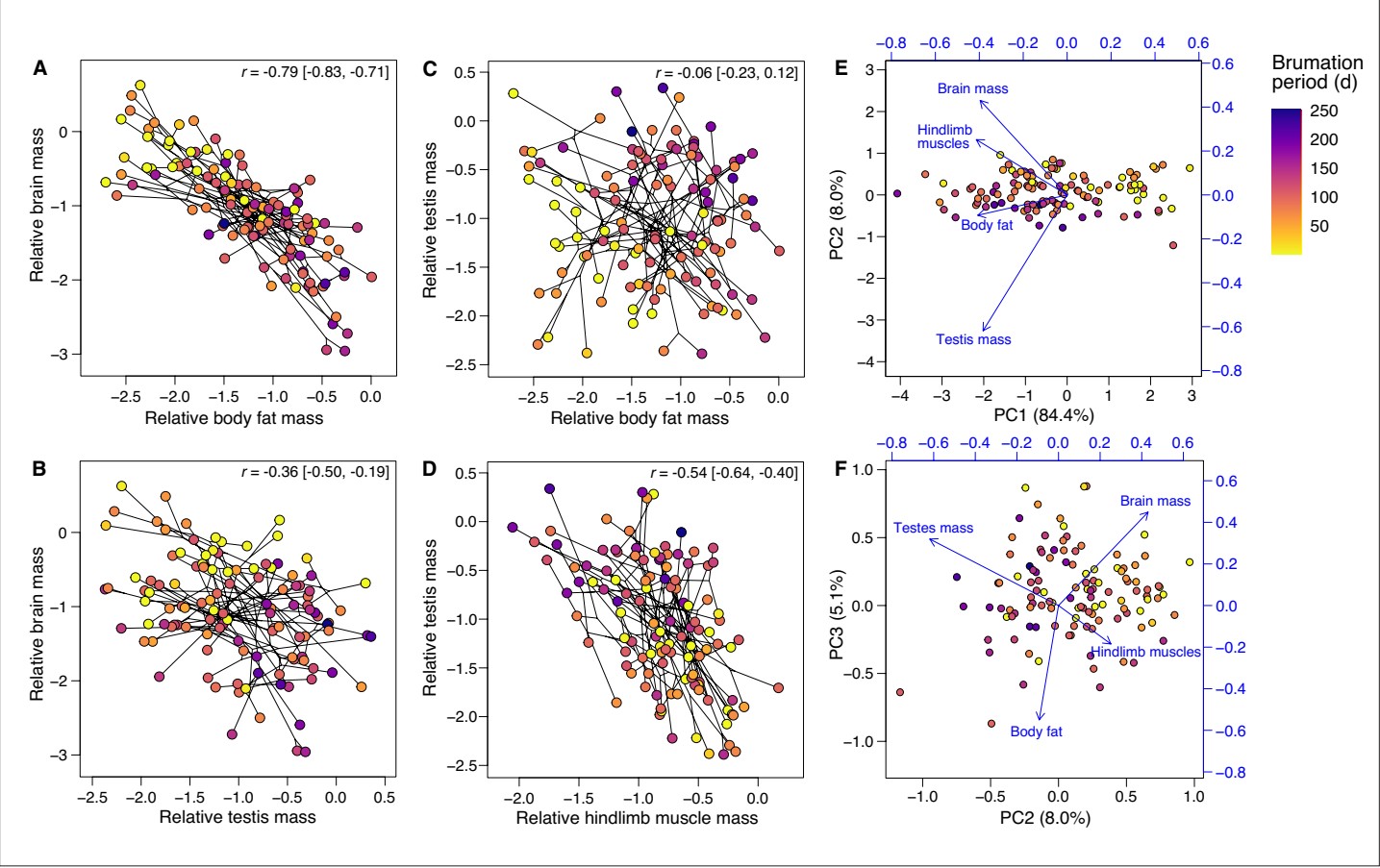

**Figure 3.** Effects of brumation duration on the relative tissue sizes. Panels (**A–D**) depict the phylogenetic correlations (shown as phylomorphospace plots; **Revell, 2012**) between the relative masses of (**A**) brain and body fat, (**B**) brain and testes, (**C**) testes and body fat, and (**D**) testes and hindlimb muscles, respectively, across the 116 species (results in **Appendix 1—table 14**). The relative tissue masses represent the centered log ratios of the compositional data, and the lines connect the nodes of the underlying phylogeny, indicating that phenotypic correlations are not simply the result of phylogenetic clustering. The correlation coefficients and 95% confidence intervals are indicated. The loadings from a phylogenetic principal component analysis (**Revell, 2012**) on the same variables are also mapped as vectors onto biplots between (**E**) the first and second or (**F**) the second and third principal components. In all panels, the point colors reflect the species-specific brumation periods (see legend in panel **A**). Generally, where brumation was relatively shorter or absent, species also tended to have relatively larger brains, less body fat and smaller testes, respectively, consistent with the univariate analyses (**Figure 2**).

mass, generating a five-variable compositional dataset (**van den Boogaart and Tolosana-Delgado, 2013**). The combined mass of all four focal tissues scaled proportionately with body size (allometric $\beta$ = 1.02 [0.96, 1.07], $\lambda$ = 0.01 [0.00, 0.11]), confirming size-independent proportion of the total resources allocated to the four focal tissues combined. However, the resource distribution among these tissues varied considerably between species. Pairwise correlations between the four focal tissues, transformed to centered log ratios (**van den Boogaart and Tolosana-Delgado, 2008**) and controlling for phylogeny, showed brain mass to covary negatively with fat and testis mass, while testis mass covaried negatively with hindlimb muscle mass but not with body fat (**Figure 3A–D**; **Appendix 1—table 14**).

To further examine the effect of brumation duration on all five variables simultaneously, we conducted a phylogenetic multivariate regression analysis (**Clavel et al., 2015**) on the same compositional data, but now transformed to isometric log ratios as recommended for multivariate models (**van den Boogaart and Tolosana-Delgado, 2013**). Brumation duration significantly affected anuran body composition (Pillai's trace = 0.33, effect size $\xi^2$ = 0.30, p = 0.001). The back-transformed coefficients of body fat (0.23) and testis mass (0.26) exceeded the expected coefficient of 0.20 if brumation had no effect, while brain mass, hindlimb muscles, and the rest of the body had lower coefficients (0.16, 0.17, and 0.18, respectively; also see **Figure 3**).

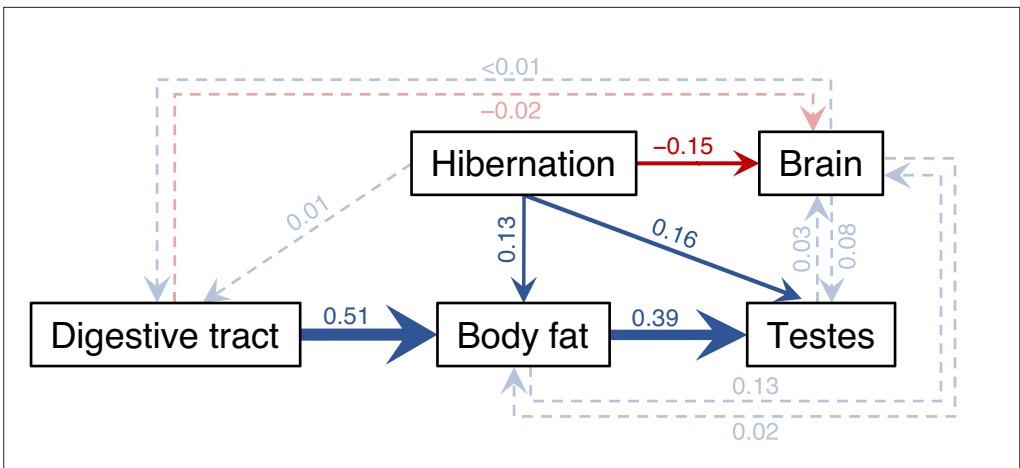

**Figure 4.** Results of the averaged phylogenetic path model. Visual representation of the average phylogenetic path model across 116 anuran species. Arrows reflect the direction of the path, with their widths being proportional to the standardized regression coefficients and colors indicating the sign (blue = positive, red = negative). Paths with 95% confidence intervals (CIs) excluding 0 (i.e., arrows highly probable) are drawn as solid arrows, all others as dashed, semi-transparent arrows. For simplicity and to avoid overparameterization, other organs were omitted in path models as they showed little covariation with brumation duration or brain size. All phenotypic traits were log-transformed, and all variables were controlled for body size via additional paths from log SVL. Although snout-vent length (SVL) had a strong effect on all variables (all $\beta > 0.37$), its thick blue arrows to each box are omitted in this figure only for visual clarity, but all path coefficients are presented with their 95% CI in **Appendix 1—figure 7**, with further details in **Appendix 1—figure 6** and **Appendix 1—table 6**.

Finally, a phylogenetically informed principal component analysis (**Revell, 2012**) confirmed the negative associations of brain size with body fat and testis mass, and that of testis mass with hind-limb muscles (**Figure 3E, F**, **Appendix 1—table 15**). Here, the first three principal components (PC1 to PC3) explained 84.7%, 8.0%, and 5.1% of total variance, respectively. Although PC2 and PC3 explained a relatively small proportion of the total variance, they separated the different tissues. PC2 loaded mainly by brain size (0.30) and testis mass (−0.44), and PC3 by brain size (0.26) and body fat (−0.31). Biplots indicated negative associations between the vectors of the same traits as above within the multivariate trait space (**Figure 3F**). Furthermore, brumation duration covaried negatively with PC2 ($r = -0.53$, $t_{114} = -6.59$, p < 0.0001, $\lambda = 0.74$ [0.49, 0.92]), consistent with reduced brain size and increased in testis mass toward longer brumation, but it was not associated with PC3 ($r = -0.12$, $t_{114} = -1.30$, p = 0.20, $\lambda = 0.41$ [0.00, 0.70]).

## Direct and indirect effects revealed by path analysis

To untangle the evolutionary links between tissue sizes, and to test competing brain evolution hypotheses in the context of brumation, we conducted a phylogenetic confirmatory path analysis (**Gonzalez-Voyer and Hardenberg, 2014**; **von Hardenberg and Gonzalez-Voyer, 2013**) with 28 predetermined path models (**Appendix 1—figure 6**; **Appendix 1—table 16**). The averaged model (**Figure 4**) confirmed the negative effect of prolonged brumation on relative brain size ($\beta = -0.15$ [−0.22, −0.07]), along with direct ($\beta = 0.16$ [0.07, 0.26]) and indirect positive effects on relative testis size (**Appendix 1—figure 7**). These effects on testis size were mediated by the relative amount of adipose tissue, which increased with brumation duration ($\beta = 0.13$ [0.06, 0.20]) and digestive tract size ($\beta = 0.51$ [0.37, 0.65]), and in turn positively influenced relative testis size ($\beta = 0.39$ [0.18, 0.59]).

## Discussion

Our study on anurans, with validated brumation periods and direct measures of expensive tissues, provides novel insights into brain and reproductive evolution in 'cold-blooded' organisms exposed to environmental seasonality. We found that species with longer brumation exhibited relatively smaller brains and allocated greater fat reserves primarily in reproduction, possibly due to the shorter breeding season with its socio-ecological implications.

## Environmental and resource considerations in brain evolution

We demonstrated that species in cooler and more seasonal climates were more cold tolerant, thereby likely optimizing their active period. Yet, low temperatures impair foraging and digestion in ectotherms (e.g., *Fontaine et al., 2018*; *Riddle, 1909*), such that high seasonality may lead to longer brumation and smaller brains. These results confirm that, unlike birds (*Sol, 2009*) and some mammals (*van Woerden et al., 2010*), challenging and unpredictable environmental conditions select for physiological rather than cognitive buffering in anurans (*Luo et al., 2017*). Supporting a large brain may not be sustainable without continued resource intake, or larger brains could be less tolerant to hypoxic conditions during brumation (*Sukhum et al., 2016*). However, selection for relatively larger brains may also simply be stronger in species with longer active (and short brumation) periods owing to extended cognitive benefits such as predator evasion (*Kotrschal et al., 2015*; *Liao et al., 2022*) or exploitation of better and more diverse food sources (*Jiang et al., 2023*; *Lefebvre et al., 1997*).

In pairwise comparisons, the relative sizes of the tissues examined here, including the brain, were generally positively correlated. These results reject both the expensive tissue and energy trade-off hypotheses (*Aiello and Wheeler, 1995*; *Isler and van Schaik, 2006*), which predict trade-offs of brain size with the size of the digestive tract or other costly organs, respectively. This lack of support in anurans aligns with a previous report in mammals (*Navarrete et al., 2011*) despite their smaller brains and vastly different ecology and physiology, including a lower metabolic rate and largely lacking physiological thermoregulation. When focusing jointly on the four tissues (brain, body fat, testes, and hindlimb muscles) that covaried with brumation duration, however, relative brain size covaried negatively with the relative mass of both fat tissue and testes, supporting the fat–brain trade-off (*Navarrete et al., 2011*) or expensive sexual tissue hypotheses (*Pitnick et al., 2006*), respectively.

## Brumation and fat tissue

Species with longer brumation periods exhibited relatively more total body fat and a higher degree of its depletion, supporting the hypothesis that anurans buffer lean periods by metabolizing stored fat (*Huang et al., 2020*; *Luo et al., 2017*). Although adipose tissue may not itself be metabolically expensive, transporting it adds costs to locomotion, particularly when jumping away from predators (*Moreno-Rueda et al., 2020*) or climbing trees compared to moving horizontally on land or in water (*Alexander, 2003*; *Hanna et al., 2008*). Consistent with this notion, arboreal species tended to be leaner compared to (semi)aquatic or terrestrial species (*Appendix 1—table 17*), controlling for brumation duration and relative brain size, both of which we had shown to covary with body fat (*Figures 2 and 3*).

## Brumation and testis evolution

Species with prolonged brumation also had relatively smaller hindleg muscles and larger testes. The negative relationship between hindleg muscle mass and brumation duration may be linked to more movement during a longer active period, including predator evasion (*Liao et al., 2022*; *Marchisin and Anderson, 1978*). Larger testes may result from a shorter breeding season, leading to denser and more synchronous mating activity (*Wells, 2007*), as suggested by our path analysis. Breeding aggregations increase male–male competition over fertilization and thus enhanced investments in sperm production (*Liao et al., 2018*; *Lüpold et al., 2020*). Our results thus reveal how brumation patterns, influenced by environmental variation and physical constraints, affect the socio-ecological context of breeding, the mode and degree of sexual selection, and ultimately the evolution of mating systems, broadening *Emlen and Oring, 1977* general predictions.

In addition to the average size of the testes, their seasonal change also varied with the brumation period. Seasonally breeding anurans regress and regrow their testes between mating seasons (*Ogielska and Bartmańska, 2009*). Non-brumating species can use energy uptake to compensate for testicular recrudescence, while those with a short breeding season after a prolonged inactive period depend on the stored fat to regrow their testes before or immediately after emergence from their hibernaculum. Hence, resources are diverted away from the brain and other organs, especially in species such as *Brachytarsophrys* spp., in which the fully developed testes combined weigh 12–14 times more than the brain *Source data 1*.

## Brain–testis trade-off

A phylogenetic path analysis confirmed the negative effect of brumation duration on relative brain size and revealed its direct and indirect positive effects on relative testis size, mediated by the amount of adipose tissue, which responded to variation in the inactive period (energetic demand) and the size of the digestive tract (energy uptake). That body fat did not contribute to brain size evolution in this more comprehensive analysis compared to pairwise correlations suggests that the fat–brain trade-off may not be direct. Rather, longer brumation, and thus a short active period, may enhance selection on fat storage for testicular investments in addition to starvation avoidance, while reducing selection for larger brains due to a shifted balance between cognitive benefits and energetic costs (*Figure 3*).

A brain–testis trade-off has been reported for bats (*Pitnick et al., 2006*), but not replicated later in the same (*Dechmann and Safi, 2009*) or other mammalian taxa (*Lemaître et al., 2009*). In anurans, the apparent trade-off may result indirectly from opposing selection on brain and testis sizes via environmental seasonality and relative durations of the active and inactive periods. The testes may evolve in response to increased sperm competition and depletion during the shorter and more synchronized breeding season. The brain, while also responding to sexual selection (*Mai et al., 2020*), is central to various activities, including feeding (*Lefebvre et al., 1997*) or predator avoidance (*Kotrschal et al., 2015*; *Liao et al., 2022*) that are themselves subject to climatic conditions and may independently influence brain evolution. Additionally, whereas testes can regress to save energy when inactive (*Ogielska and Bartmańska, 2009*), brain metabolism may be less reducible (*Mink et al., 1981*), resulting in different cost–benefit balances between these organs in relation to seasonality.

## Conclusions

In conclusion, our analyses show how brumation in anurans, influenced by high environmental seasonality, may affect resource allocation between costly tissues, directly or through its environmental correlates. The non-independent selective processes promoting diversification in different traits emphasize the need to study the evolutionary trajectory of a given trait such as brain size in the immediate context of both simultaneous investments to other tissues and the species-specific ecology. Our findings also have important implications in the context of sustained environmental change, exacerbated by climate change with its effects on temperature and precipitation patterns around the globe (*IPCC, 2022*). Such climatic shifts may disrupt the timing of brumation and breeding in many anurans, change their exposure to pathogens or predators through modified activity patterns, affect their ability to find food or suitable breeding sites, or change their population dynamics through resource availability, intra- and interspecific competition, dispersal capabilities, and gene flow (*Alves-Ferreira et al., 2022*; *Blaustein et al., 2010*; *Blaustein et al., 2001*; *Carey and Alexander, 2003*). In species with temperature-dependent sexual differentiation, thermal shifts may further change operational sex ratios in breeding populations and thus likely mating dynamics and sexual selection (*Eggert, 2004*; *Lüpold et al., 2017*; *Ruiz-García et al., 2021*). All these factors impose intense environmental pressure on resource acquisition and allocation patterns, and it remains to be seen to what extent variation in the adaptability, and thus resilience, between species exposed to environmental change is attributable to such competing needs between investments and species-specific constraints.

# Materials and methods
## Sample collection and preparation

Between 2010 and 2020 and as part of concurrent studies, we collected a total of 396 sexually mature males from 116 anuran species (3.41 ± 0.95 males each) in post-brumation breeding condition and an additional 132 adult males from 50 of these species (2.64 ± 0.94 males each) shortly before entering their hibernacula (*Source data 1* and *Source data 2*). For each species, we sampled all males at a single location in southern and western China with known longitude, latitude, and elevation (*Source data 3*). Upon transfer to the laboratory, we sacrificed the individuals by single-pithing, measured their SVL to the nearest 0.01 mm with calipers and then preserved them in 4% phosphate-buffered formalin for tissue fixation.

After 2 months of preservation, we weighed each complete specimen to the nearest 0.1 mg using an electronic balance to obtain body mass before dissecting them following a strict protocol. We separately extracted the brain, heart, liver, lungs, kidneys, spleen, digestive tract, testes, limb muscles,

and fat stores, cleaned these tissues and immediately weighed them to the nearest 0.1 mg with an electronic balance. We additionally measured the length of the digestive tract to the nearest 0.01 mm using calipers. We excluded emaciated individuals or those exhibiting visible organ pathologies from our analyses.

## Environmental seasonality

For each collection site, we retrieved from the 30-year climate history of https://www.meteoblue.com the monthly mean temperature (in °C) and total precipitation (in mm) (*Source data 3*) and used these values to calculate location-specific annual means and coefficients of variation. We also determined the duration of the dry season, P2T, as the number of months, for which the total precipitation was less than twice the mean temperature (*Walter, 1971*).

## Brumation period

One way that anurans can physiologically respond to seasonality is by adjusting their thermal sensitivity and thus brumation period (*Wells, 2007*), which in turn could directly or indirectly affect the evolution of brain size (*Heldstab et al., 2018*). Hence, we estimated the brumation period for all 116 species. To this end, we visited the field sites for 30 of our species daily around the expected start and end times of brumation (based on prior experience). For each species, we recorded the dates and temperatures (using a Kobold HND-T105 high-precision thermometer to the nearest 0.1) when the last frogs of a given species were seen at the end of their active period (with no further activity detected for at least 7 days) and when the first individuals were detected in the spring. For the same 30 species (and using the same individuals as for morphological measurements), we then experimentally simulated brumation using a Q18 temperature-controlled refrigerator in Shenzhen Pioneer (SAST). We gradually lowered and raised the temperature at a rate of 0.5°C/hr and recorded the temperature at which test subjects entered and left brumation. When entering brumation, anurans drop their heart rate, become sluggish, draw their nictitating membranes across the eyes for protection, spread their legs for stability, and change their physiology to avoid freezing, start breathing through their skin, or switch to anaerobic metabolism (*Fei and Ye, 2001*; *Pinder et al., 1992*; *Tattersall and Ultsch, 2008*; *Wells, 2007*). To minimize disturbance in our experiment, we used the motion-less four-point stance with the nictitating membranes drawn across the eyes as our proxy of brumation. Our experimental threshold temperatures were tightly associated with the corresponding field measurements both for the start ($r = 0.97$, $t_{28} = 22.26$, $p < 0.0001$, $\lambda = 0.04$ [0.00, 0.47]) and end of the inactive state ($r = 0.98$, $t_{28} = 28.05$, $p < 0.0001$, $\lambda = 0.04$ [0.00, 0.43]). Hence, we assessed the corresponding temperatures for all remaining species in the laboratory and estimated the brumation period based on the daily mean temperatures at the corresponding collection sites as retrieved from Chinese Meteorological Stations (http://www.lishi.tianqi.com) between 2012 and 2016.

We defined the brumation period as the number of consecutive days in each year that remained below this threshold. For simplicity, we determined the active rather than brumation period, starting with the first day that the mean daily temperature rose above the activity threshold and remained there for at least five consecutive days, and ending with the last day before the temperature dropped below the activity threshold and remained there until the end of the calendar year. The brumation period then represented the difference between the activity period and the total number of days in each calendar year. Across these 5 years, the measured temperature thresholds yielded highly repeatable species-specific estimates of the number of days below the activity range ($R = 0.95$ [95% CI: 0.93–0.96]), as determined by the *rpt* function in the *rptR* package (*Stoffel et al., 2017*) across all 116 species (*Appendix 1—figure 1*; *Appendix 1—table 1*). Furthermore, across the 30 species that were examined both in the lab and the field (see above), these predicted brumation periods were also correlated with the observed brumation periods in the field ($r = 0.96$, $t_{28} = 1\,8.03$, $p < 0.0001$, $\lambda = 0.05$ [0.00, 0.48]; *Appendix 1—figure 8A*), which themselves were highly repeatable between years within species ($R = 0.98$ [0.96–0.99]; *Appendix 1—table 1*).

Based on this data validation, we used for each species the mean brumation period predicted from our experimentally simulated temperature thresholds. However, to test for potential buffering effects of burrowing in the soil relative to the air temperatures reported by the meteorological stations, we also repeated these estimates by using more conservative thermal thresholds. Here, we restricted the putative brumation days to those with a reported air temperature of either 2 or

4°C below the experimentally derived inactivity thresholds, simulating prolonged activity by seeking shelter in burrows. The 2°C threshold was based on a pilot study comparing direct measurements of air and burrow temperatures for four different burrows in each of five of our study species (burrow depths: 32.0 ± 3.2 to 121.0 ± 17.8 cm; *Appendix 1—figure 9*). Across these species, the burrow-to-air temperature difference reached 1.03 ± 0.35 to 2.45 ± 0.60°C in measurements around the peak of the brumation period (i.e., early January; Figure S9). However, since these temporal snapshots were based on sites at relatively low elevation (≤320 m a.s.l.) due to accessibility of burrows during winter, we also used a second, more conservative buffer (4°C below activity range) for comparison. These temperature buffers shortened the predicted brumation periods to a varying degree between species (*Appendix 1—figure 8*); yet the predicted periods covaried strongly between the different temperature thresholds (all $r > 0.90$, $t_{114} > 21.96$, $p < 0.0001$, all $\lambda < 0.01$).

## Phylogeny reconstruction

To reconstruct the phylogeny, we obtained the sequences of three nuclear and six mitochondrial genes from GenBank (for accession numbers and sequence coverage see *Source data 4*). The three nuclear genes included the recombination-activating gene 1 (RAG1), rhodopsin (RHOD), and tyrosinase (TYR). The six mitochondrial genes were cytochrome *b* (CYTB), cytochrome oxidase subunit I (COI), NADH dehydrogenase subunits 2 and 4 (ND2 and ND4), and the large and small subunits of the mitochondrial ribosome genes (12S/16S; omitting the adjacent tRNAs as they were difficult to align and represented only a small amount of data). We aligned the sequences by multi-sequence alignment (MUSCLE) in MEGA v.10.2.2 (*Tamura et al., 2013*) before comparing possible nucleotide substitution models. The best substitution model, as determined by the function *modelTest*() in the R (*R Development Core Team, 2022*) package *phangorn* (*Schliep, 2011*) based on the corrected Akaike information criterion, AICc, was GTR + $\Gamma$ + I for all genes except RHOD, for which HKY + $\Gamma$ had stronger support.

Using BEAUTi and BEAST v.1.10.4 (*Suchard et al., 2018*), we then constructed the phylogeny with unlinked substitution models, a relaxed uncorrelated log-normal clock, a Yule speciation process, and the best-supported nucleotide substitution models. We omitted time calibration due to a lack of fossil dates. We ran the Markov Chain Monte Carlo simulation for 55 million generations while sampling every 5000th tree with a 10% burn-in. Most effective sample size values by far exceeded 375 (i.e., all well above the recommended threshold of 200) for all but two tree statistics in the program Tracer v.1.7.2 (*Rambaut et al., 2018*), thus indicating satisfying convergence of the Bayesian chain and adequate model mixing. Finally, we generated a maximum clade credibility tree with mean node heights and a 10% burn-in using TreeAnnotator v.1.10.4 (*Suchard et al., 2018*), presented in *Appendix 1—figure 10*.

## Breeding conditions

To test if a prolonged brumation period reduces the time available for reproduction, thereby changing the level of competition over mates and fertilizations (*Lüpold et al., 2017*), we extracted the start and end dates of the breeding season from our field notes of concurrent studies on species-specific life histories. These data were available for 43 of our species (*Source data 3*). We used dates when the first and last clutches were observed in focal ponds as a proxy of mating activity, given that males release their sperm during oviposition in these external fertilizers. For each species, dates from at least 2 years were combined and averaged to obtain the mean duration of the breeding season.

We further recorded whether dense mating aggregations are typically observed in these species. We have previously shown that larger mating clusters, with multiple males clasping the same females, have a significant effect on the evolution of testis size due to the resulting competition among sperm for fertilization (*Lüpold et al., 2017*). Here, we had no detailed data on the sizes of aggregations and so were only able to code the typical presence or absence of aggregations as a binary variable (*Source data 3*).

Finally, we used our direct estimates of species-specific population densities from our previous study (*Lüpold et al., 2017*) to test whether a shorter breeding season results in denser breeding populations. Although population density is a more direct measure than the occurrence of aggregations, such data were available for only eight of our species, each based on multiple populations per species (*Lüpold et al., 2017*). All these data were not necessarily derived from the same years or populations

of our main dataset, but given the within-species repeatability in breeding populations (*Lüpold et al., 2017*) and in the duration of the breeding season (R = 0.88 [0.79–0.93]; *Appendix 1—table 1*), these differences should be relatively small compared to the interspecific variation and mostly introduce random noise.

## Data analyses

### General methods

We conducted all statistical analyses in R v.4.2.0 (*R Development Core Team, 2022*), using log-transformed data for all phenotypic traits, and for the coefficient of variance (CV) in temperature among the ecological variables. To account for non-independence of data due to common ancestry (*Freckleton et al., 2002*; *Pagel, 1999*), we conducted PGLS or phylogenetic logistic regressions (e.g., for occurrence of breeding aggregations), using the R package *phylolm* (*Ho and Ané, 2014*) and our reconstructed phylogeny. To account for variation around the species means, we bootstrapped for each model (at 100 fitted replicates) the standardized regression coefficients along with the phylogenetic scaling parameter $\lambda$ and calculated their corresponding 95% CIs. The $\lambda$ values indicate phylogenetic independence near zero and strong phylogenetic dependence near one (*Freckleton et al., 2002*).

Unless stated otherwise, all PGLS models focusing on the relative mass of tissues as the response included SVL as a covariate in addition to the focal predictor variable(s). We chose SVL instead of body mass because it is the commonly used measure of body size in anurans and independent of seasonal fluctuations in tissues such as body fat, testes, or limb muscles. One exception, however, was the analysis of phylogenetically informed allometric relationships, for which we cubed SVL such that a slope of 1 equaled unity (isometry). For these allometric relationships we calculated ordinary (generalized) least-squares rather than reduced major-axis regressions, because their greater sensitivity to changes in the steepness, but lower sensitivity to changes in scatter, capture allometric slopes more adequately (*Kilmer and Rodríguez, 2017*).

### Pairwise correlations between tissues

To examine the covariation between different tissues across species, we first calculated pairwise partial correlations controlling for SVL and phylogeny. To this end, we calculated the phylogenetic trait variance–covariance matrix between the pairs of focal variable and SVL using the function *phyl.vcv*() in *phytools* (*Revell, 2012*) with $\lambda = 1$ (i.e., Brownian motion), which we then scaled into a correlation matrix using *cov2cor*() in the *stats* package (*R Development Core Team, 2022*). Using the resulting correlation coefficients $r_{xy}$, $r_{xz}$, and $r_{yz}$, respectively, we then calculated the partial correlation coefficient $r_{xy.z}$ between the x and y variables of interest while accounting for SVL (z) following *Crawley, 2007* equation: $r_{xy.z} = \frac{r_{xy} - r_{xz}r_{yz}}{\sqrt{\left(1-r_{xz}^2\right)\left(1-r_{yz}^2\right)}}$ , with the associated t-statistics and 95% CIs converted using standard conversion ($t = r\sqrt{\frac{df}{1-r^2}}$) and the package *effectsize* (*Ben-Shachar et al., 2020*), respectively.

### Multivariate allocation patterns

Since pairwise correlations do not necessarily capture more complex, multivariate allocation patterns, we used two additional approaches to explore how tissue sizes varied relative to others: A compositional analysis and a principal component analysis. In both analyses, we focused on those four tissues that covaried with brumation duration or deviated from proportionate scaling with body size: brain, body fat, testes, and hindlimb muscles.

### Compositional analyses

In the first approach, we used the function *acomp*() in the R package *compositions* (*van den Boogaart and Tolosana-Delgado, 2008*) to partition total body mass of each species into a five-variable Aitchison composition in a logistic geometry (*van den Boogaart and Tolosana-Delgado, 2013*), consisting of the proportional representation of the four focal tissues and the remaining body mass combined. Since the focal tissues constituted a size-independent fraction of the total body, the closed composition of this combined mass should be unbiased relative to body size but can instead reveal differential contributions of the four tissues to their total in a multivariate context (*Aitchison, 1982*; *Muldowney*

*et al., 2001*; *van den Boogaart and Tolosana-Delgado, 2013*). For phylogenetic correlations between these variables following the description above, we used centered log ratios obtained by the function *clr*() in the same package, which maintains the original variable structure. However, owing to the reliance on a full rank of the covariance in multivariate analyses, we used the *ilr*() function to project the $D$-part composition isometrically to a $D − 1$ dimensional simplex (*Aitchison, 1982*), essentially representing the log ratios between the $D$ parts. This multivariate object we subjected to a phylogenetic multivariate regression against brumation duration using the functions *mvgls*() and *manova.gls*() in the package *mvMORPH* (*Clavel et al., 2015*). For interpretation in the context of the original variable space, we back-transformed the coefficients using the *ilrInv*() function in *compositions*.

## Phylogenetic principal component analysis

In addition to this compositional data analysis, we also peerformed a phylogenetically informed principal component analysis on the same focal tissues as log-transformed species means, using the *phyl. pca*() function of the package *phytools* (*Revell, 2012*). Here, we primarily focused on the directions of the loading vectors relative to the principal components and one another to glean information on the correlations between the original variables in the principal component space.

## Phylogenetic confirmatory path analyses

We performed two phylogenetic confirmatory path analyses (*Gonzalez-Voyer and Hardenberg, 2014*; *von Hardenberg and Gonzalez-Voyer, 2013*) based on pre-specified candidate structural equation models. In the first analysis, we explored the direct and indirect effects of climatic variables on the duration of the breeding season or formation of breeding aggregations. Using eight candidate structural equation models, we essentially tested if breeding aggregations were the result of variation in latitude as a proxy of seasonality, either directly or mediated by the duration of brumation and/ or of the breeding season. To avoid overparameterization given that this analysis was based on only 43 species, we did not include testis mass (and so necessarily also SVL to control for body size) as additional variables in the same path models. Rather, to test for links between relative testis size and breeding parameters, we conducted separate directional tests of trait evolution (see below).

A second path analysis aimed to disentangle the different interrelationships between traits that could ultimately mediate the effect of brumation duration on brain and reproductive evolution. Here, we used 28 pre-specified structural equation models to test for direct and indirect links between the brumation period and the four main tissues. As brumation is determined primarily by the environment, we focused on models that explained variation in tissue investments rather than, say, brain size affecting brumation patterns. The effects of brumation on the four tissues were direct or indirect, for example mediated by the digestive tract (resource acquisition) and/or body fat (resource storage), thus providing context for the cognitive buffer, expensive brain, expensive tissue, energy trade-off, and fat−brain trade-off hypotheses (*Allman et al., 1993*; *Heldstab et al., 2016*; *Isler and van Schaik, 2006*; *Isler and van Schaik, 2009*; *Navarrete et al., 2011*). Furthermore, we allowed brain size to affect testis size and vice versa (i.e., expensive sexual tissue hypothesis; *Pitnick et al., 2006*), and included models in which these organs explained variation in the digestive tract or fat tissue instead of being affected by them (e.g., selection on brain or testis size might mediate selection for greater resource availability rather than resources influencing brain or testis evolution). Across the 28 candidate models, we tested different combinations of these predictions, with traits being explained by single or multiple predictors, or having individual or shared effects on other traits. Using the R package *phylopath* (*van der Bijl, 2018*), we examined the conditional independencies of each model, ranked all candidate models based on their $C$-statistic information criterion (CICc), and then averaged the coefficients of the models with ΔCICc ≤2 from the top model (*von Hardenberg and Gonzalez-Voyer, 2013*).

## Directional tests of trait evolution

Since the first path analysis did not involve testis size to avoid overparametization, we separately tested for correlated evolution using directional tests of trait evolution (*Pagel, 1994*; *Revell, 2012*). One limitation of these models is that they rely on evolutionary transitions between binary states in each trait. Hence, we considered positive residuals of a log–log regression between testis mass and SVL as 'relatively large testes' and negative residuals as 'relatively small testes'. For the duration of

the breeding season, we similarly split the distribution based on the mean duration, whereas aggregation formation was already coded as present or absent. Based on (the weight of) the AIC, we then tested if changes in relative testis size and breeding parameters, respectively, were unilaterally dependent, mutually dependent, or independent (*Pagel, 1994*), using the *fitPagel*() function in the *phytools* package (*Revell, 2012*) with 'fitDiscrete' as the optimization method and allowing all rates to differ (i.e., 'ARD' model).

## Acknowledgements

We thank K Isler and C van Schaik for helpful comments on an earlier draft of our manuscript and J Santin for insightful discussions. Financial support was provided by the National Natural Sciences Foundation of China (32370456, 31970393 and 31772451 to WBL), the Key Project of Natural Science Foundation of Sichuan Province (22NSFSC0011 to WBL) and by the Swiss National Science Foundation (PP00P3_170669 and PP00P3_202662 to SL).

## Additional information

### Funding

| Funder | Grant reference number | Author |
| --- | --- | --- |
| National Natural Science Foundation of China | 32370456 | Wenbo Liao |
| National Natural Science Foundation of China | 31970393 | Wenbo Liao |
| National Natural Science Foundation of China | 31772451 | Wenbo Liao |
| Natural Science Foundation of Sichuan Province | 22NSFSC0011 | Wenbo Liao |
| Swiss National Science Foundation | PP00P3_170669 | Stefan Lüpold |
| Swiss National Science Foundation | PP00P3_202662 | Stefan Lüpold |

The funders had no role in study design, data collection, and interpretation, or the decision to submit the work for publication.

### Author contributions

Wenbo Liao, Conceptualization, Validation, Writing – original draft, Writing – review and editing; Ying Jiang, Resources, Data curation; Long Jin, Investigation, Methodology; Stefan Lüpold, Formal analysis, Writing – original draft

### Author ORCIDs

Wenbo Liao ⓘ https://orcid.org/0000-0001-5303-4114
Ying Jiang ⓘ http://orcid.org/0000-0002-6391-5537
Stefan Lüpold ⓘ http://orcid.org/0000-0002-5069-1992

Joint Public Review: https://doi.org/10.7554/eLife.88236.3.sa1
Author Response https://doi.org/10.7554/eLife.88236.3.sa2

## Additional files

### Supplementary files
• MDAR checklist

• Source data 1. Data related to environmental conditions, brumation, and breeding.

• Source data 2. Morphological data of male anurans in breeding (post-brumation) condition. Snout-vent length (SVL), body mass and the size of all relevant tissues and organs. N = sample size.

• Source data 3. Morphological data of male anurans in pre-brumation condition. Snout-vent length (SVL), body mass and the size of all relevant tissues and organs. N = sample size.

• Source data 4. Genbank accession numbers used for phylogeny construction.

• Source data 5. Monthly temperatures and precipitation corresponding to the location of each species from 2012 to 2016, based on daily climate data.

### Data availability

Source data and associated codes are available at figshare https://doi.org/10.6084/m9.figshare.21078052. In the source data, annual mean temperature and total precipitation are retrieved from https://www.meteoblue.com (*Source data 1*), and daily mean temperatures are retrieved from http://www.lishi.tianqi.com (*Source data 5*).

The following dataset was generated:

| Author(s) | Year | Dataset title | Dataset URL | Database and Identifier |
| --- | --- | --- | --- | --- |
| Lüpold S, Liao WB, Jiang Y, Jin L | 2023 | Data from: How hibernation in frogs drives brain and reproductive evolution in opposite directions | https://doi.org/10.6084/m9.figshare.21078052 | figshare, 10.6084/m9.figshare.21078052 |

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

# Appendix 1

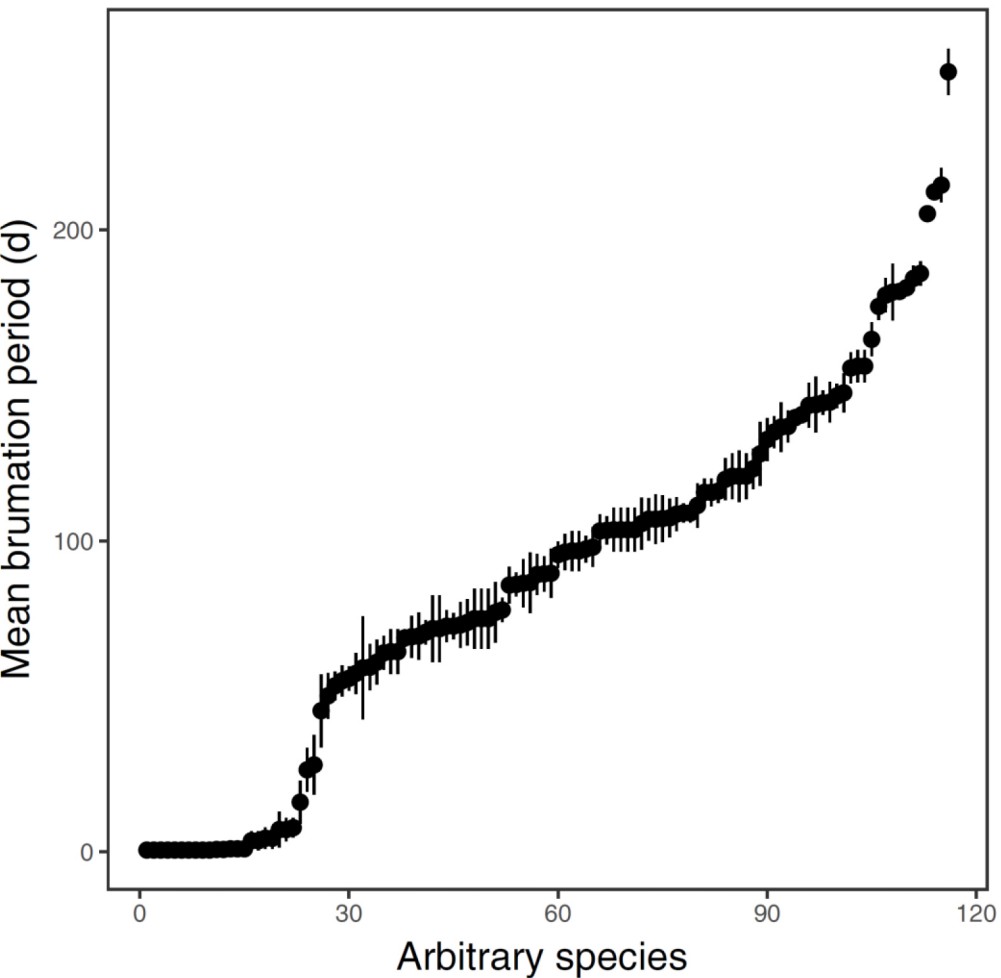

**Appendix 1—figure 1.** Distribution of brumation durations (in days) across the 116 anuran species. Each dot represents the species-specific mean ± 1 SE of the 5 years (2012–2016) that specimens were collected. For each year and species, all days with a collection site-specific ambient temperature below the experimentally quantified, species-specific temperature thresholds were summed to estimate the brumation period.

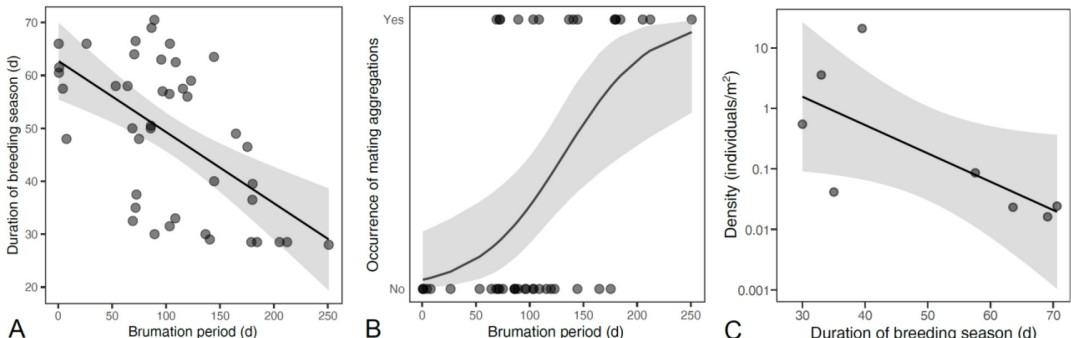

**Appendix 1—figure 2.** Effects of brumation on breeding conditions. Prolonged brumation shortened the breeding season (**A**), thereby increasing the probability of dense breeding aggregations (**B**). This effect was also supported by a trend toward higher mean population densities in species with a shorter breeding season (**C**).

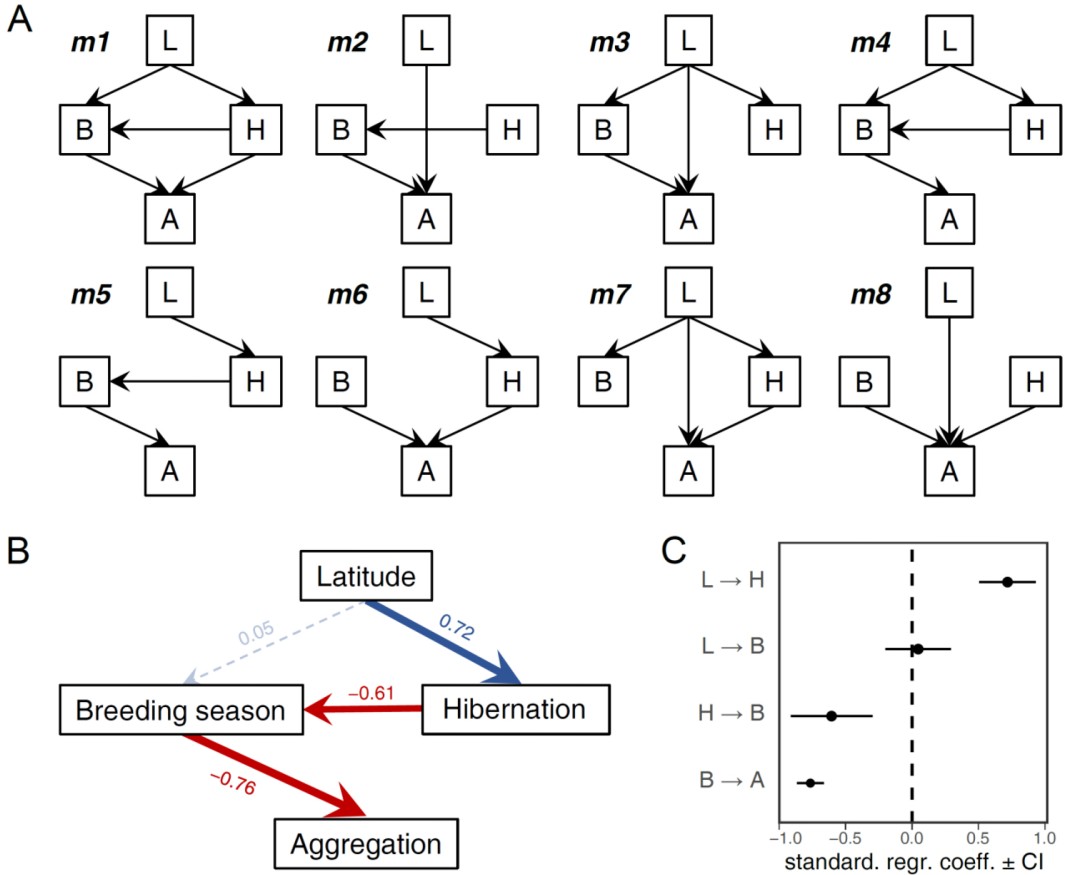

**Appendix 1—figure 3.** Confirmatory path analysis on the links between environmental variation, the brumation duration and breeding activity and the formation of breeding aggregations. Directed acyclic graphs representing 8 candidate models (**A**) formed the predictions for the path analysis, resulting in the average path model (**B**). The path coefficients (standardised regression coefficients) and their 95% confidence intervals are depicted in (**C**). The variable letters correspond to the first letters of the full variable names in panel B. The model comparisons are listed in *Appendix 1—table 13*. The results were qualitatively identical when substituting latitude with elevation, mean temperature, or the coefficient of variation in temperature (except that the effect of mean temperature on brumation duration was negative as predicted). Note that here we used 'hibernation' instead of 'brumation' to avoid conflicts in codes.

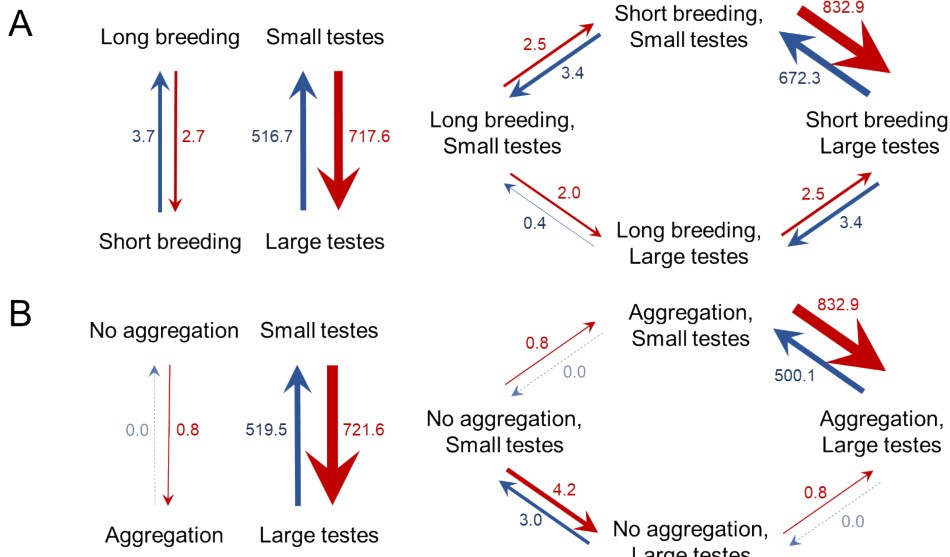

**Appendix 1—figure 4.** Directional test of trait evolution between relative testis size and breeding conditions. Transition rates between binary states for relative testis size and (**A**) the duration of the breeding season or (**B**) aggregation formation. Independent models are shown on the left, models with changes in relative testis size dependent on the breeding parameters are depicted on the right. For both pairs of variables, the independent model had the highest support (**A**: wAIC = 0.50, **B**: wAIC = 0.66), but the dependent models shown here were either not significantly less supported (**A**: ΔAIC = 0.83, wAIC = 0.33) or relatively weakly so (**B**: ΔAIC = 2.38, wAIC = 0.20). The remaining two models (changes in breeding parameters depending on those in relative testis size or both variables evolving interdependently) found much less support in both analyses (**A**: ΔAIC ≥ 3.40, wAIC ≤ 0.09, **B**: ΔAIC ≥ 3.64, wAIC ≤ 0.11).

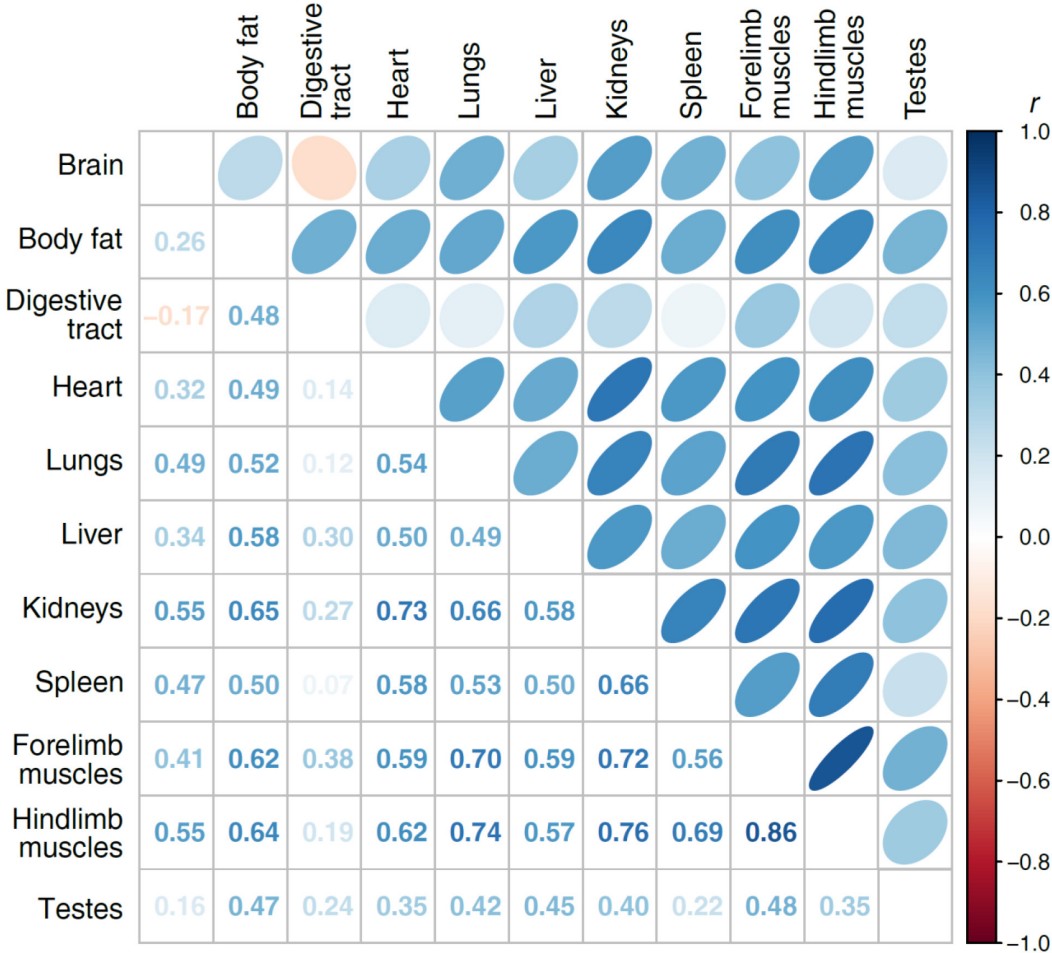

**Appendix 1—figure 5.** Pairwise partial correlations between the different tissues. All correlations are controlled for snout-vent length and phylogeny, expressed both by ellipses (above diagonal) and the correlation coefficients (below the diagonal). All correlations with |rp| ≥ 0.19 had P < 0.05 before and after accounting for multiple testing based on the False Discovery Rate.

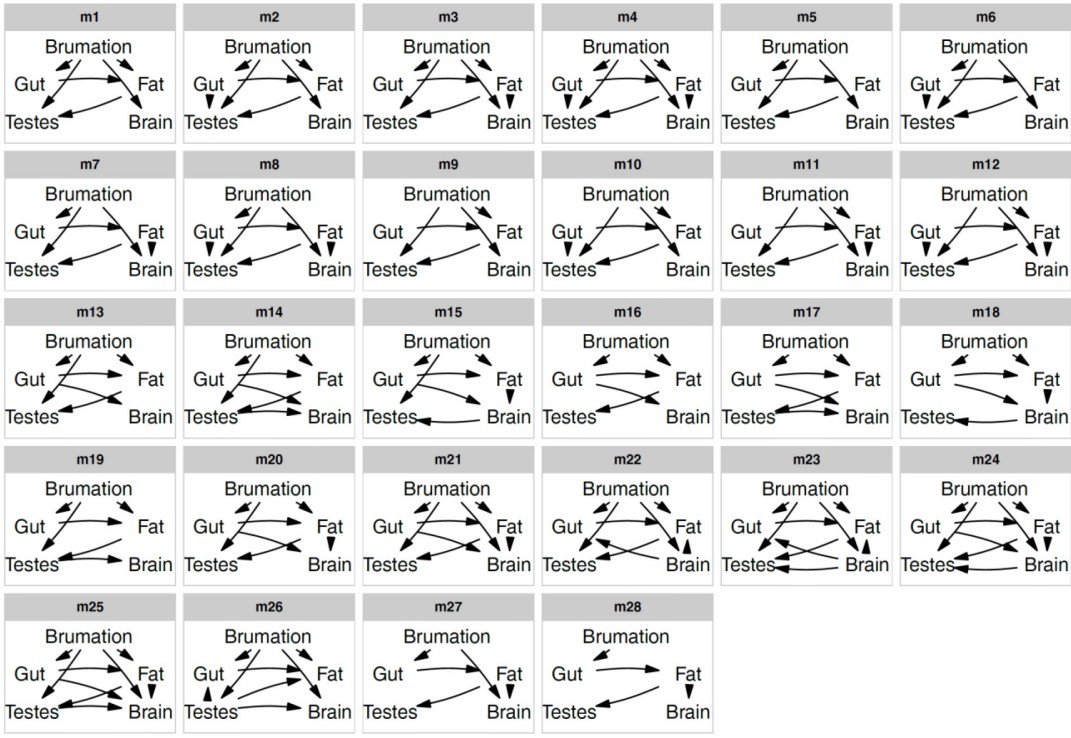

**Appendix 1—figure 6.** Candidate path models. Directed acyclic graphs representing 28 candidate models that were compared to disentangle the relationships between five traits through a phylogenetic confirmatory path analysis and multi-model inference. For spatial reasons, the digestive tract is termed gut, and we omitted the paths from snout-vent length to all five variables in each panel to control tissue masses for body size or to improve d-separation (in the case of brumation).

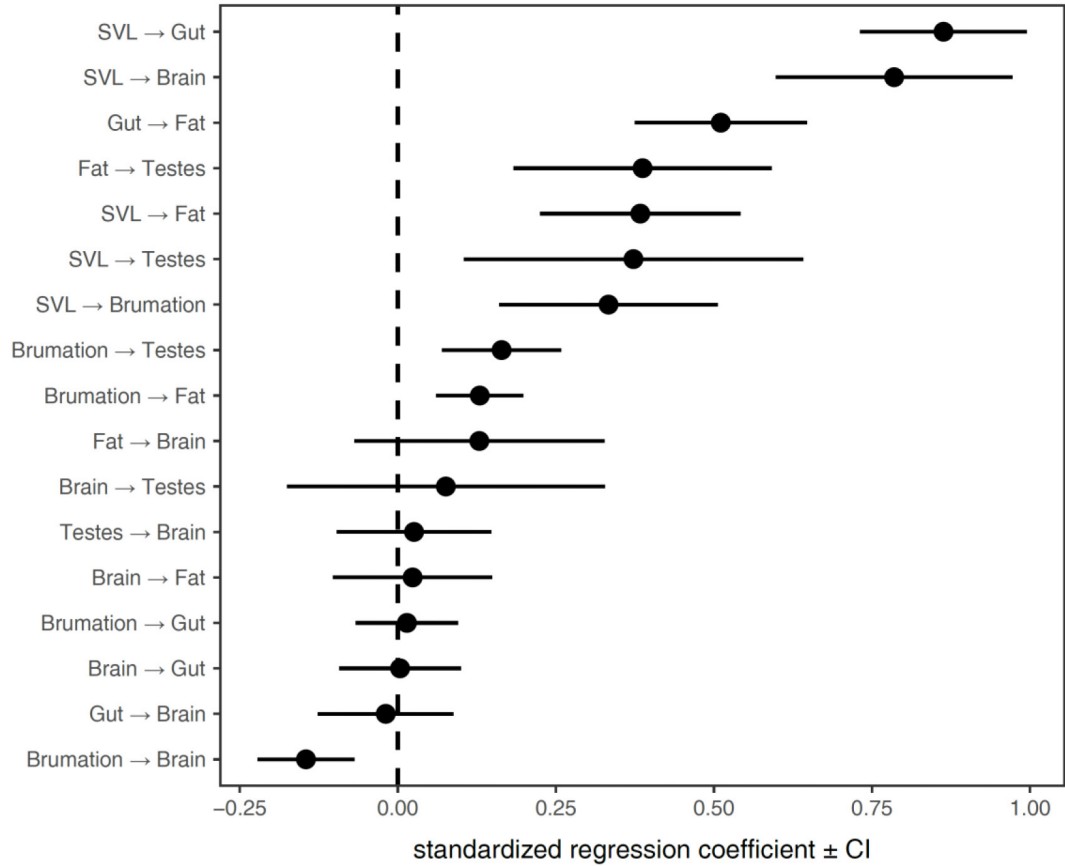

**Appendix 1—figure 7.** Coefficients of the averaged phylogenetic path model. The path coefficients (standardised regression coefficients) are depicted with their 95% confidence intervals, corresponding to the path analysis in *Figure 4* (directed path models in *Appendix 1—figure 6*). The paths from SVL to other traits are included only to control for body size (in the case of tissues) or to improve d-separation (SVL → Brumation). For visual clarity, these are omitted in the directed acyclic graphs (*Appendix 1—figure 6*) and the final averaged model (*Figure 4*).

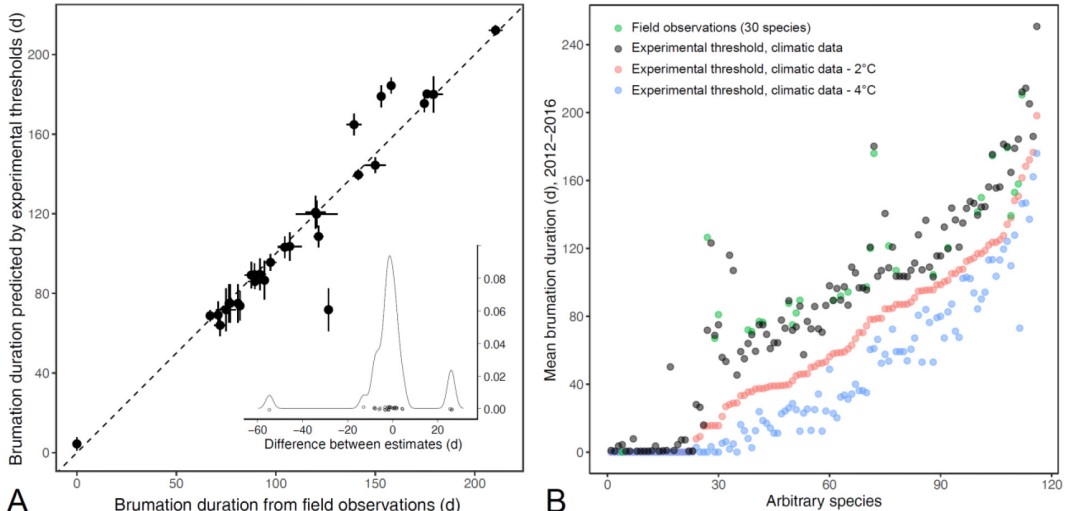

**Appendix 1—figure 8.** Validation of the different estimates of brumation duration. (**A**) Comparisons of field observations with predicted brumation periods based on experimental thermal thresholds and site-specific climate data. The dashed line represents unity. The inserted density plot indicates that twenty-five of the 30
*Appendix 1—figure 8 continued on next page*

*Appendix 1—figure 8 continued*

estimated values were within ≤8 days of the field observations across the spectrum, with the remaining five estimates deviating by approx. 13, 26 (3×) or, for one outlier, 55 days. (**B**) Variation in species-specific brumation periods based on different approaches. The green dots are the same field-based data as in panel A. The black dots represent values that combine the experimental temperature thresholds with the site-specific climatic data (predicted brumation below the threshold). The red and blue dots are based on the same experimental thresholds, but this time following the air temperature profiles of 2 and 4°C below these thresholds, respectively, to account for the potential buffering effect of the soil for underground hibernacula. Since the green dots were most closely associated with the black ones (i.e., without the additional buffer), we focused primarily on those values, but conducted additional analyses using the shorter, buffered brumation periods with no qualitative change. Finally, the horizontal dashed line splits the species into those with brumation likely present (above) or absent (below) to generate binary brumation variables for more direct comparison with previous studies in other taxa. This cut-off of ≤40 versus >40 days was motivated by the relatively large gap in the sequence of brumation periods in both the black and blue dots and that anurans are likely to survive relatively short cold spells without special adaptations. This cut-off generated divergent datasets between the direct experimental thresholds (black: N = 91 species with and 25 without prolonged brumation) and buffered thresholds (blue: N = 47 species with and 69 without prolonged brumation).

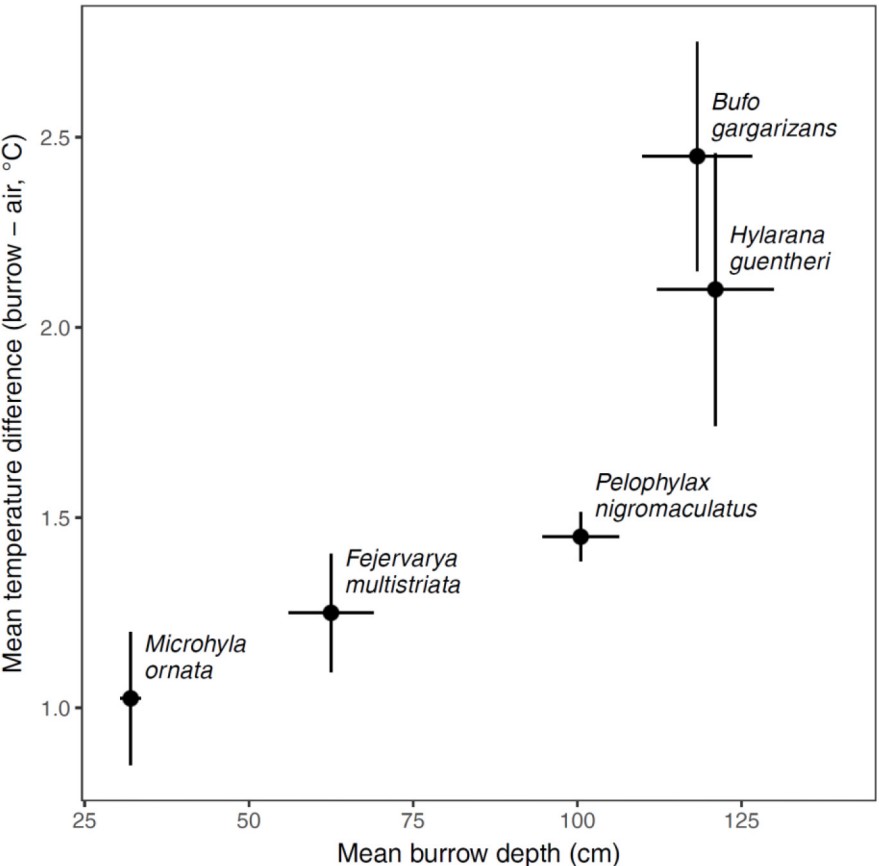

**Appendix 1—figure 9.** Comparison of air and burrow temperatures. The difference between burrow and outside air temperature increased with burrow depth across the five species examined in a pilot study. The points reflect the means with standard errors around them based on four burrows per species.

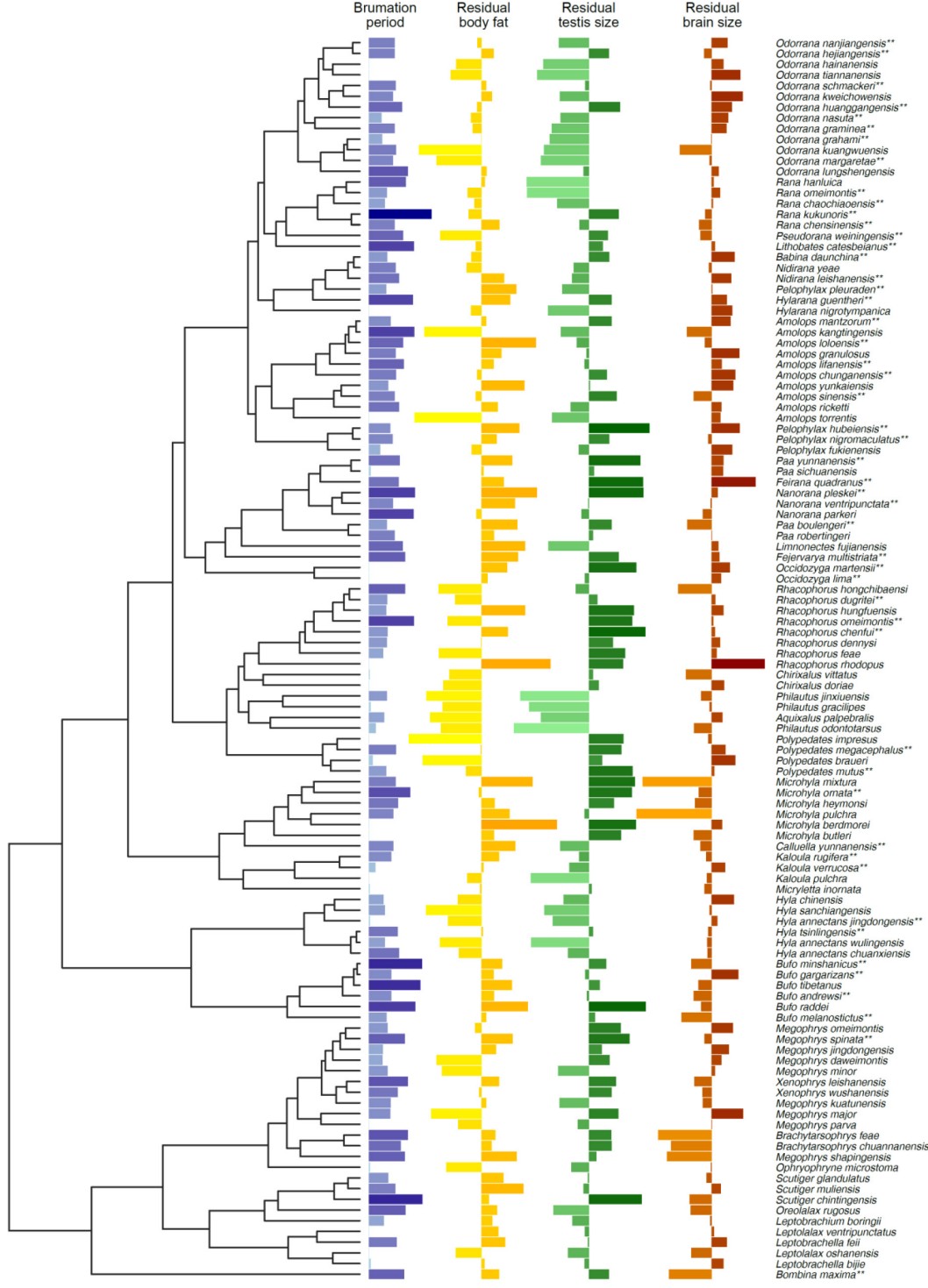

**Appendix 1—figure 10.** Phylogeny of the species used in the study. Species with data on pre-brumation condition are labelled with double-asterisks (**). Residual trait sizes are derived from log-log linear regressions against snout-vent length.

**Appendix 1—table 1.** Within-species repeatability in male morphology, climatic variables, and brumation duration.

CV = coefficient of variation, P2T = duration of the dry season (in months), with a month defined as dry when its total precipitation is less than two times the mean temperature. All analyses are based

on the males (in breeding condition) or specific sampling locations for N = 116 species, except field observations of brumation duration (N = 30 species).

| Variable | N | R [95% CI] | P |
|---|---|---|---|
| Male tissue masses | | | |
| SVL | 396 | 0.950 [0.933, 0.962] | <0.001 |
| Brain | 396 | 0.948 [0.930, 0.961] | <0.001 |
| Body fat | 396 | 0.958 [0.942, 0.969] | <0.001 |
| Digestive tract | 396 | 0.949 [0.931, 0.962] | <0.001 |
| Heart | 396 | 0.918 [0.890, 0.939] | <0.001 |
| Lungs | 396 | 0.920 [0.890, 0.939] | <0.001 |
| Liver | 396 | 0.915 [0.885, 0.936] | <0.001 |
| Kidneys | 396 | 0.921 [0.893, 0.941] | <0.001 |
| Spleen | 396 | 0.937 [0.914, 0.952] | <0.001 |
| Forelimb muscles | 396 | 0.947 [0.929, 0.960] | <0.001 |
| Hindlimb muscles | 396 | 0.961 [0.946, 0.971] | <0.001 |
| Testes | 396 | 0.947 [0.929, 0.961] | <0.001 |
| Climate variables | | | |
| Mean temperature | 580 | 0.982 [0.976, 0.986] | <0.001 |
| Mean precipitation | 580 | 0.757 [0.693, 0.802] | <0.001 |
| CV temperature | 580 | 0.978 [0.971, 0.982] | <0.001 |
| CV precipitation | 580 | 0.401 [0.327, 0.486] | <0.001 |
| P2T | 580 | 0.688 [0.614, 0.751] | <0.001 |
| Brumation | | | |
| Brumation duration (estimated from lab data) | 580 | 0.947 [0.928, 0.959] | <0.001 |
| Brumation duration (field observations) | 66 | 0.982 [0.965, 0.991] | <0.001 |
| Breeding season | | | |
| Breeding season duration | 86 | 0.881 [0.785, 0.936] | <0.001 |

**Appendix 1—table 2.** Species-specific environmental effects on the brumation duration. CV = coefficient of variation, P2T = duration of the dry season (in months), with a month defined as dry when its total precipitation is less than two times the mean temperature. All analyses are based on df = 114, and confidence intervals are bootstrapped (N = 100 simulations). $\lambda$ = phylogenetic scaling parameter.

| Predictor | β [95% CI] | r [95% CI] | t | P | λ [95% CI] |
|---|---|---|---|---|---|
| Latitude | 0.72 [ 0.61, 0.82] | 0.72 [ 0.63, 0.78] | 11.03 | <0.001 | 0.00 [0.00, 0.10] |
| Longitude | 0.09 [-0.06, 0.28] | 0.09 [-0.09, 0.27] | 1.00 | 0.321 | 0.47 [0.00, 0.69] |
| Elevation | 0.43 [ 0.24, 0.62] | 0.43 [ 0.28, 0.56] | 5.15 | <0.001 | 0.00 [0.00, 0.10] |
| Mean temperature | −0.56 [-0.69,−0.38] | −0.56 [-0.66,−0.42] | −7.14 | <0.001 | 0.00 [0.00, 0.11] |
| Mean precipitation | −0.30 [-0.45,−0.13] | −0.30 [-0.45,−0.13] | −3.42 | <0.001 | 0.00 [0.00, 0.10] |
| CV temperature | 0.65 [ 0.51, 0.78] | 0.65 [ 0.54, 0.73] | 9.08 | <0.001 | 0.00 [0.00, 0.11] |
| CV precipitation | 0.01 [-0.18, 0.21] | 0.01 [-0.17, 0.19] | 0.06 | 0.948 | 0.42 [0.00, 0.65] |
| P2T | −0.37 [-0.52,−0.19] | −0.38 [-0.51,−0.21] | −4.38 | <0.001 | 0.42 [0.00, 0.66] |

**Appendix 1—table 3.** Species-specific environmental effects on the brumation duration, excluding those 25 species that are unlikely to hibernate (≤27 days).

CV = coefficient of variation, P2T = duration of the dry season (in months), with a month defined as dry when its total precipitation is less than two times the mean temperature. All analyses are based on df = 89, and confidence intervals are bootstrapped (N = 100 simulations). $\lambda$ = phylogenetic scaling parameter.

| Predictor | β [95% CI] | r [95% CI] | t | P | λ [95% CI] |
|---|---|---|---|---|---|
| Latitude | 0.46 [ 0.28, 0.65] | 0.46 [ 0.29, 0.60] | 4.94 | <0.001 | 0.00 [0.00, 0.11] |
| Longitude | −0.15 [-0.35, 0.08] | −0.15 [-0.33, 0.06] | −1.38 | 0.170 | 0.00 [0.00, 0.07] |
| Elevation | 0.42 [ 0.21, 0.61] | 0.42 [ 0.24, 0.56] | 4.38 | <0.001 | 0.00 [0.00, 0.11] |
| Mean temperature | −0.36 [-0.58,–0.15] | −0.36 [-0.52,–0.17] | −3.69 | <0.001 | 0.00 [0.00, 0.10] |
| Mean precipitation | −0.35 [-0.51,–0.12] | −0.35 [-0.51,–0.16] | −3.54 | <0.001 | 0.00 [0.00, 0.08] |
| CV temperature | 1.17 [ 0.47, 1.80] | 0.72 [ 0.28, 0.87] | 3.42 | 0.006 | 0.79 [0.00, 1.00] |
| CV precipitation | 0.03 [-0.16, 0.25] | 0.03 [-0.18, 0.23] | 0.27 | 0.785 | 0.00 [0.00, 0.06] |
| P2T | −0.21 [-0.38,–0.01] | −0.21 [-0.39, 0.00] | −2.02 | 0.046 | 0.00 [0.00, 0.09] |

**Appendix 1—table 4.** Species-specific environmental effects on the temperature at which males entered or left their inactive state.

All analyses are limited to those 91 species that are likely to overwinter (≥47 days below experimental temperature threshold), with df = 89, and confidence intervals are bootstrapped (N = 100 simulations). CV = coefficient of variation, P2T = duration of the dry season (in months), with a month defined as dry when its total precipitation is less than two times the mean temperature, $\lambda$ = phylogenetic scaling parameter.

| Predictor | β [95% CI] | r [95% CI] | t | P | λ |
|---|---|---|---|---|---|
| Entering inactive state: | | | | | |
| Latitude | −0.10 [-0.26, 0.09] | −0.12 [-0.31, 0.09] | −1.16 | 0.247 | 0.32 [0.00, 0.56] |
| Longitude | 0.24 [ 0.09, 0.41] | 0.31 [ 0.11, 0.47] | 3.02 | 0.003 | 0.00 [0.00, 0.07] |
| Elevation | −1.42 [-1.94,–0.99] | −0.51 [-0.63,–0.34] | −5.56 | <0.001 | 0.07 [0.00, 0.23] |
| Mean temperature | 0.24 [ 0.13, 0.35] | 0.46 [ 0.28, 0.59] | 4.84 | <0.001 | 0.00 [0.00, 0.10] |
| Mean precipitation | 0.01 [-0.01, 0.03] | 0.07 [-0.13, 0.27] | 0.71 | 0.480 | 0.39 [0.00, 0.61] |
| CV temperature | −1.78 [-2.69,–0.88] | −0.39 [-0.54,–0.20] | −4.03 | <0.001 | 0.00 [0.00, 0.07] |
| CV precipitation | −3.62 [-6.87, 0.10] | −0.21 [-0.39, 0.00] | −1.99 | 0.049 | 0.00 [0.00, 0.06] |
| P2T | 0.35 [ 0.01, 0.63] | 0.22 [ 0.02, 0.40] | 2.18 | 0.032 | 0.43 [0.00, 0.66] |
| Leaving inactive state: | | | | | |
| Latitude | −0.12 [-0.30, 0.08] | −0.14 [-0.33, 0.07] | −1.34 | 0.184 | 0.47 [0.00, 0.68] |
| Longitude | 0.28 [ 0.12, 0.46] | 0.35 [ 0.15, 0.50] | 3.48 | <0.001 | 0.37 [0.00, 0.59] |
| Elevation | −1.63 [-2.13,–1.18] | −0.55 [-0.66,–0.39] | −6.23 | <0.001 | 0.23 [0.00, 0.46] |
| Mean temperature | 0.21 [ 0.09, 0.32] | 0.39 [ 0.20, 0.54] | 3.97 | <0.001 | 0.35 [0.00, 0.59] |
| Mean precipitation | 0.01 [ 0.00, 0.03] | 0.14 [-0.07, 0.33] | 1.32 | 0.191 | 0.47 [0.00, 0.68] |
| CV temperature | −1.33 [-2.30,–0.39] | −0.30 [-0.46,–0.10] | −2.93 | 0.004 | 0.44 [0.00, 0.66] |
| CV precipitation | −4.43 [-8.14,–0.72] | −0.25 [-0.42,–0.05] | −2.43 | 0.017 | 0.42 [0.00, 0.64] |
| P2T | 0.22 [-0.13, 0.53] | 0.14 [-0.07, 0.33] | 1.33 | 0.188 | 0.53 [0.00, 0.73] |

**Appendix 1—table 5.** Associations between species-specific environmental effects and male body size.

Associations of either snout-vent length or body mass as two measures of body size with brumation

duration and different environmental variables. All analyses based on df = 114, and confidence intervals are bootstrapped (N = 100 simulations). CV = coefficient of variation, P2T = duration of the dry season (in months), with a month defined as dry when its total precipitation is less than two times the mean temperature, and $\lambda$ = phylogenetic scaling parameter.

| Predictor | β [95% CI] | r [95% CI] | t | P | λ [95% CI] |
|---|---|---|---|---|---|
| Snout-vent length: | | | | | |
| Brumation duration | 0.02 [-0.04, 0.07] | 0.06 [-0.13, 0.23] | 0.59 | 0.557 | 0.93 [0.82, 0.99] |
| Latitude | 0.04 [-0.03, 0.09] | 0.13 [-0.06, 0.30] | 1.36 | 0.177 | 0.93 [0.82, 0.99] |
| Elevation | –0.01 [-0.07, 0.03] | –0.05 [-0.23, 0.13] | –0.53 | 0.597 | 0.94 [0.85, 1.00] |
| Mean temperature | –0.01 [-0.05, 0.05] | –0.04 [-0.21, 0.15] | –0.38 | 0.702 | 0.93 [0.82, 0.99] |
| CV temperature | 0.01 [-0.05, 0.06] | 0.04 [-0.14, 0.22] | 0.42 | 0.676 | 0.93 [0.82, 0.99] |
| P2T | 0.01 [-0.02, 0.06] | 0.05 [-0.13, 0.23] | 0.55 | 0.583 | 0.94 [0.83, 0.99] |
| Body mass: | | | | | |
| Brumation duration | –0.03 [-0.23, 0.14] | –0.03 [-0.21, 0.15] | –0.36 | 0.722 | 0.92 [0.80, 0.98] |
| Latitude | 0.10 [-0.12, 0.27] | 0.09 [-0.09, 0.27] | 1.00 | 0.321 | 0.91 [0.77, 0.97] |
| Elevation | –0.16 [-0.32,–0.02] | –0.18 [-0.34, 0.01] | –1.91 | 0.058 | 0.94 [0.85, 1.00] |
| Mean temperature | 0.03 [-0.11, 0.23] | 0.03 [-0.15, 0.21] | 0.37 | 0.710 | 0.92 [0.79, 0.99] |
| CV temperature | –0.03 [-0.22, 0.12] | –0.03 [-0.21, 0.15] | –0.36 | 0.721 | 0.92 [0.79, 0.99] |
| P2T | 0.14 [0.02, 0.29] | 0.16 [-0.02, 0.33] | 1.72 | 0.087 | 0.93 [0.81, 0.98] |

**Appendix 1—table 6.** Associations between tissue size and body size.
Log-log association of tissue mass with snout-vent length. To facilitate the interpretation of the allometric slopes (β), SVL is cubed (SVL$^3$) so that β = 1 equals isometry. All analyses are based on df = 114, and confidence intervals are bootstrapped (N = 100 simulations). $\lambda$ = phylogenetic scaling parameter.

| Response | β [95% CI] | r [95% CI] | t | P | λ [95% CI] |
|---|---|---|---|---|---|
| Brain | 0.49 [0.44, 0.54] | 0.88 [0.84, 0.91] | 19.66 | <0.001 | 0.73 [0.35, 0.87] |
| Body fat | 1.36 [1.28, 1.48] | 0.94 [0.92, 0.95] | 29.50 | <0.001 | 0.39 [0.00, 0.64] |
| Digestive tract | 0.95 [0.88, 1.01] | 0.91 [0.88, 0.93] | 23.60 | <0.001 | 0.00 [0.00, 0.10] |
| Heart | 1.07 [0.98, 1.17] | 0.89 [0.86, 0.92] | 21.27 | <0.001 | 0.18 [0.00, 0.45] |
| Lungs | 1.06 [0.96, 1.19] | 0.87 [0.83, 0.90] | 18.81 | <0.001 | 0.23 [0.00, 0.50] |
| Liver | 1.03 [0.96, 1.10] | 0.93 [0.91, 0.95] | 27.12 | <0.001 | 0.00 [0.00, 0.10] |
| Kidneys | 1.03 [0.97, 1.09] | 0.95 [0.93, 0.96] | 30.90 | <0.001 | 0.00 [0.00, 0.10] |
| Spleen | 1.10 [0.98, 1.26] | 0.84 [0.79, 0.88] | 16.56 | <0.001 | 0.38 [0.00, 0.64] |
| Forelimb muscles | 1.23 [1.16, 1.30] | 0.94 [0.93, 0.96] | 30.62 | <0.001 | 0.00 [0.00, 0.10] |
| Hindlimb muscles | 1.10 [1.03, 1.18] | 0.95 [0.93, 0.96] | 31.01 | <0.001 | 0.49 [0.00, 0.71] |
| Testes | 1.06 [0.94, 1.22] | 0.82 [0.76, 0.86] | 15.27 | <0.001 | 0.56 [0.00, 0.77] |

**Appendix 1—table 7.** Results of phylogenetically controlled relationships ($\lambda$ = phylogenetic scaling parameter) between the brumation duration and the relative size of different male tissues in post-brumation breeding (N = 116 species) or pre-brumation condition (N = 50 species), measured as tissue mass.
All associations are controlled for snout-vent length (SVL), and confidence intervals are bootstrapped (N = 100 simulations).

*Appendix 1—table 7 Continued on next page*

| Response | Predictors | β [95% CI] | Partial r [95% CI] | t | P | λ [95% CI] |
|---|---|---|---|---|---|---|
| **Males in breeding condition (df = 113)** | | | | | | |
| Brain | Brumation | −0.09 [-0.15,−0.04] | −0.31 [-0.46,−0.14] | −3.50 | <0.001 | 0.35 [0.00, 0.61] |
| | SVL | 0.65 [ 0.59, 0.71] | 0.91 [ 0.87, 0.93] | 22.76 | <0.001 | |
| Body fat | Brumation | 0.20 [0.04, 0.34] | 0.25 [0.07, 0.40] | 2.72 | 0.008 | 0.42 [0.00, 0.66] |
| | SVL | 1.55 [1.37, 1.73] | 0.87 [0.83, 0.90] | 18.72 | <0.001 | |
| Digestive tract | Brumation | 0.03 [-0.11, 0.14] | 0.04 [-0.14, 0.22] | 0.45 | 0.656 | 0.00 [0.00, 0.11] |
| | SVL | 1.12 [ 1.01, 1.25] | 0.86 [ 0.81, 0.89] | 17.66 | <0.001 | |
| Heart | Brumation | −0.07 [-0.22, 0.06] | −0.09 [-0.26, 0.09] | −0.96 | 0.340 | 0.23 [0.00, 0.51] |
| | SVL | 1.35 [ 1.20, 1.50] | 0.87 [ 0.82, 0.90] | 18.49 | <0.001 | |
| Lungs | Brumation | −0.05 [-0.23, 0.09] | −0.06 [-0.24, 0.12] | −0.69 | 0.490 | 0.21 [0.00, 0.50] |
| | SVL | 1.31 [ 1.15, 1.49] | 0.83 [ 0.77, 0.87] | 15.86 | <0.001 | |
| Liver | Brumation | 0.05 [-0.09, 0.16] | 0.07 [-0.11, 0.25] | 0.77 | 0.445 | 0.00 [0.00, 0.11] |
| | SVL | 1.23 [ 1.11, 1.36] | 0.87 [ 0.83, 0.90] | 19.12 | <0.001 | |
| Kidneys | Brumation | −0.02 [-0.14, 0.08] | −0.03 [-0.21, 0.15] | −0.37 | 0.715 | 0.00 [0.00, 0.11] |
| | SVL | 1.25 [ 1.15, 1.36] | 0.90 [ 0.87, 0.92] | 22.15 | <0.001 | |
| Spleen | Brumation | −0.12 [-0.31, 0.04] | −0.12 [-0.30, 0.06] | −1.32 | 0.189 | 0.23 [0.00, 0.52] |
| | SVL | 1.34 [ 1.15, 1.55] | 0.80 [ 0.74, 0.85] | 14.27 | <0.001 | |
| Foreleg muscles | Brumation | −0.06 [-0.20, 0.05] | −0.09 [-0.27, 0.09] | −0.98 | 0.329 | 0.00 [0.00, 0.11] |
| | SVL | 1.52 [ 1.40, 1.66] | 0.91 [ 0.88, 0.93] | 23.29 | <0.001 | |
| Hindleg muscles | Brumation | −0.13 [-0.25,−0.03] | −0.22 [-0.38,−0.04] | −2.37 | 0.019 | 0.22 [0.00, 0.51] |
| | SVL | 1.35 [ 1.23, 1.48] | 0.91 [ 0.88, 0.93] | 22.98 | <0.001 | |
| Testes | Brumation | 0.30 [0.15, 0.44] | 0.36 [0.19, 0.50] | 4.06 | <0.001 | 0.77 [0.40, 0.90] |
| | SVL | 1.21 [1.01, 1.41] | 0.78 [0.71, 0.83] | 13.30 | <0.001 | |
| **Males in pre-brumation condition (df = 47)** | | | | | | |
| Brain | Brumation | −0.15 [-0.21,−0.09] | −0.51 [-0.67,−0.27] | −4.03 | <0.001 | 0.00 [0.00, 0.56] |
| | SVL | 0.60 [ 0.55, 0.66] | 0.92 [ 0.88, 0.95] | 16.30 | <0.001 | |
| Body fat | Brumation | 0.33 [0.21, 0.46] | 0.53 [0.29, 0.68] | 4.26 | <0.001 | 0.55 [0.00, 0.88] |
| | SVL | 1.21 [1.04, 1.40] | 0.89 [0.82, 0.92] | 13.21 | <0.001 | |
| Digestive tract | Brumation | 0.08 [-0.04, 0.20] | 0.15 [-0.14, 0.40] | 1.01 | 0.320 | 0.64 [0.00, 0.92] |
| | SVL | 1.16 [ 0.99, 1.33] | 0.88 [ 0.81, 0.92] | 12.85 | <0.001 | |
| Heart | Brumation | −0.02 [-0.15, 0.11] | −0.03 [-0.30, 0.25] | −0.22 | 0.825 | 0.24 [0.00, 0.74] |
| | SVL | 0.89 [ 0.74, 1.07] | 0.83 [ 0.73, 0.88] | 10.11 | <0.001 | |
| Lungs | Brumation | 0.02 [-0.14, 0.18] | 0.03 [-0.24, 0.31] | 0.24 | 0.812 | 0.55 [0.00, 0.89] |
| | SVL | 1.03 [ 0.81, 1.26] | 0.80 [ 0.68, 0.86] | 9.02 | <0.001 | |
| Liver | Brumation | 0.11 [-0.02, 0.23] | 0.19 [-0.09, 0.44] | 1.34 | 0.186 | 0.00 [0.00, 0.56] |
| | SVL | 1.16 [ 1.05, 1.30] | 0.90 [ 0.85, 0.93] | 14.56 | <0.001 | |
| Kidneys | Brumation | −0.06 [-0.17, 0.04] | −0.14 [-0.39, 0.15] | −0.97 | 0.338 | 0.00 [0.00, 0.56] |
| | SVL | 0.99 [ 0.90, 1.11] | 0.91 [ 0.85, 0.94] | 14.85 | <0.001 | |
| Spleen | Brumation | 0.11 [-0.10, 0.32] | 0.12 [-0.16, 0.38] | 0.83 | 0.410 | 0.00 [0.00, 0.56] |

| Response | Predictors | β [95% CI] | Partial r [95% CI] | t | P | λ [95% CI] |
|---|---|---|---|---|---|---|
| | SVL | 1.11 [ 0.93, 1.35] | 0.77 [ 0.64, 0.85] | 8.35 | <0.001 | |
| Foreleg muscles | Brumation | –0.07 [-0.18, 0.05] | –0.14 [-0.39, 0.15] | –0.94 | 0.354 | 0.00 [0.00, 0.56] |
| | SVL | 1.32 [ 1.22, 1.45] | 0.93 [ 0.90, 0.96] | 18.00 | <0.001 | |
| Hindleg muscles | Brumation | –0.09 [-0.18, 0.01] | –0.21 [-0.45, 0.07] | –1.49 | 0.142 | 0.57 [0.00, 0.89] |
| | SVL | 1.13 [ 1.01, 1.27] | 0.93 [ 0.88, 0.95] | 16.96 | <0.001 | |
| Testes | Brumation | 0.02 [-0.14, 0.22] | 0.02 [-0.26, 0.29] | 0.15 | 0.880 | 0.82 [0.15, 0.98] |
| | SVL | 1.00 [ 0.72, 1.24] | 0.73 [ 0.58, 0.82] | 7.34 | <0.001 | |

**Appendix 1—table 8.** Results of phylogenetically controlled relationships ($\lambda$ = phylogenetic scaling parameter) of brumation on the relative mass of male tissues (N = 116 species).

For justification of assignment of brumating and non-brumating species, see *Appendix 1—figure 7*. All associations are controlled for snout-vent length (SVL), and confidence intervals are bootstrapped (N = 100 simulations).

| Response | Predictors | β [95% CI] | Partial r [95% CI] | t | P | λ [95% CI] |
|---|---|---|---|---|---|---|
| Binary classification of brumation (N=91 brumating, 25 non-brumating species; total df = 113) | | | | | | |
| Brain | Brumation | –0.12 [-0.25, 0.03] | –0.17 [-0.34, 0.01] | –1.86 | 0.066 | 0.34 [0.00, 0.60] |
| | SVL | 0.64 [ 0.58, 0.71] | 0.90 [ 0.86, 0.92] | 21.56 | <0.001 | |
| Body fat | Brumation | 0.53 [0.21, 0.93] | 0.26 [0.08, 0.42] | 2.91 | 0.004 | 0.50 [0.00, 0.72] |
| | SVL | 1.53 [1.34, 1.71] | 0.86 [0.82, 0.89] | 18.26 | <0.001 | |
| Digestive tract | Brumation | 0.01 [-0.30, 0.36] | 0.00 [-0.18, 0.19] | 0.04 | 0.965 | 0.00 [0.00, 0.10] |
| | SVL | 1.13 [ 1.02, 1.27] | 0.86 [ 0.81, 0.89] | 17.64 | <0.001 | |
| Heart | Brumation | –0.22 [-0.56, 0.16] | –0.12 [-0.30, 0.06] | –1.34 | 0.184 | 0.21 [0.00, 0.49] |
| | SVL | 1.36 [ 1.21, 1.51] | 0.87 [ 0.82, 0.90] | 18.61 | <0.001 | |
| Lungs | Brumation | –0.07 [-0.47, 0.36] | –0.04 [-0.22, 0.15] | –0.39 | 0.697 | 0.20 [0.00, 0.48] |
| | SVL | 1.31 [ 1.14, 1.48] | 0.83 [ 0.77, 0.87] | 15.71 | <0.001 | |
| Liver | Brumation | 0.03 [-0.28, 0.39] | 0.02 [-0.16, 0.20] | 0.21 | 0.833 | 0.00 [0.00, 0.10] |
| | SVL | 1.24 [ 1.13, 1.38] | 0.87 [ 0.83, 0.90] | 19.11 | <0.001 | |
| Kidneys | Brumation | –0.11 [-0.38, 0.20] | –0.08 [-0.25, 0.11] | –0.83 | 0.408 | 0.00 [0.00, 0.10] |
| | SVL | 1.26 [ 1.16, 1.38] | 0.90 [ 0.87, 0.92] | 22.20 | <0.001 | |
| Spleen | Brumation | –0.39 [-0.83, 0.10] | –0.17 [-0.34, 0.01] | –1.83 | 0.071 | 0.23 [0.00, 0.51] |
| | SVL | 1.36 [ 1.17, 1.55] | 0.81 [ 0.74, 0.85] | 14.47 | <0.001 | |
| Foreleg muscles | Brumation | –0.07 [-0.39, 0.29] | –0.04 [-0.22, 0.14] | –0.45 | 0.653 | 0.00 [0.00, 0.10] |
| | SVL | 1.51 [ 1.39, 1.66] | 0.91 [ 0.88, 0.93] | 22.87 | <0.001 | |
| Hindleg muscles | Brumation | –0.15 [-0.44, 0.16] | –0.11 [-0.28, 0.08] | –1.12 | 0.264 | 0.17 [0.00, 0.45] |
| | SVL | 1.34 [ 1.22, 1.44] | 0.90 [ 0.87, 0.93] | 22.41 | <0.001 | |
| Testes | Brumation | 0.49 [ 0.13, 0.89] | 0.23 [ 0.05, 0.39] | 2.49 | 0.014 | 0.75 [0.36, 0.90] |
| | SVL | 1.23 [ 1.02, 1.43] | 0.77 [ 0.70, 0.82] | 12.91 | <0.001 | |
| Binary classification of brumation with a 4 °C buffer (N=47 brumating, 69 non-brumating species; total df = 113) | | | | | | |
| Brain | Brumation | –0.21 [-0.31,–0.14] | –0.38 [-0.51,–0.21] | –4.33 | <0.001 | 0.35 [0.00, 0.62] |
| | SVL | 0.64 [ 0.59, 0.71] | 0.91 [ 0.88, 0.93] | 23.47 | <0.001 | |

*Appendix 1—table 8 Continued on next page*

*Appendix 1—table 8 Continued*

| Response | Predictors | β [95% CI] | Partial r [95% CI] | t | P | λ [95% CI] |
|---|---|---|---|---|---|---|
| Body fat | Brumation | 0.29 [ 0.00, 0.52] | 0.19 [ 0.01, 0.36] | 2.07 | 0.041 | 0.46 [0.00, 0.72] |
| | SVL | 1.57 [ 1.39, 1.77] | 0.87 [ 0.83, 0.90] | 18.88 | <0.001 | |
| Digestive tract | Brumation | 0.07 [-0.17, 0.23] | 0.06 [-0.13, 0.23] | 0.60 | 0.553 | 0.00 [0.00, 0.11] |
| | SVL | 1.12 [ 1.01, 1.24] | 0.87 [ 0.82, 0.90] | 18.43 | <0.001 | |
| Heart | Brumation | −0.25 [-0.52,−0.07] | −0.18 [-0.35, 0.00] | −1.97 | 0.051 | 0.23 [0.00, 0.51] |
| | SVL | 1.35 [ 1.21, 1.52] | 0.88 [ 0.83, 0.90] | 19.23 | <0.001 | |
| Lungs | Brumation | −0.31 [-0.61,−0.10] | −0.20 [-0.36,−0.01] | −2.13 | 0.035 | 0.21 [0.00, 0.50] |
| | SVL | 1.32 [ 1.17, 1.51] | 0.84 [ 0.79, 0.88] | 16.68 | <0.001 | |
| Liver | Brumation | −0.09 [-0.34, 0.06] | −0.07 [-0.25, 0.11] | −0.76 | 0.45 | 0.00 [0.00, 0.11] |
| | SVL | 1.25 [ 1.14, 1.38] | 0.89 [ 0.85, 0.91] | 20.31 | <0.001 | |
| Kidneys | Brumation | −0.06 [-0.27, 0.08] | −0.05 [-0.23, 0.13] | −0.55 | 0.586 | 0.00 [0.00, 0.11] |
| | SVL | 1.24 [ 1.15, 1.35] | 0.91 [ 0.88, 0.93] | 23.04 | <0.001 | |
| Spleen | Brumation | −0.19 [-0.54, 0.05] | −0.10 [-0.28, 0.08] | −1.12 | 0.265 | 0.23 [0.00, 0.51] |
| | SVL | 1.33 [ 1.14, 1.54] | 0.80 [ 0.74, 0.85] | 14.42 | <0.001 | |
| Foreleg muscles | Brumation | −0.26 [-0.50,−0.10] | −0.19 [-0.35,−0.01] | −2.04 | 0.043 | 0.00 [0.00, 0.11] |
| | SVL | 1.52 [ 1.41, 1.65] | 0.92 [ 0.89, 0.94] | 24.62 | <0.001 | |
| Hindleg muscles | Brumation | −0.23 [-0.45,−0.08] | −0.20 [-0.36,−0.02] | −2.16 | 0.033 | 0.24 [0.00, 0.52] |
| | SVL | 1.33 [ 1.22, 1.47] | 0.91 [ 0.88, 0.93] | 23.04 | <0.001 | |
| Testes | Brumation | 0.38 [ 0.09, 0.60] | 0.24 [ 0.06, 0.40] | 2.63 | 0.01 | 0.74 [0.34, 0.89] |
| | SVL | 1.25 [ 1.03, 1.44] | 0.78 [ 0.71, 0.83] | 13.31 | <0.001 | |

**Appendix 1—table 9.** Effects of the brumation duration on relative tissue size in males, excluding those 25 species that are unlikely to overwinter (≤27 days).

Results of phylogenetically controlled relationships ( λ = phylogenetic scaling parameter) between the brumation duration and the relative mass of different male tissues in breeding condition (df = 88). All associations are controlled for snout-vent length (SVL), and confidence intervals are bootstrapped (N = 100 simulations).

| Response | Predictors | β [95% CI] | Partial r [95% CI] | t | P | λ [95% CI] |
|---|---|---|---|---|---|---|
| Brain | Brumation | −0.07 [-0.13,−0.02] | −0.27 [-0.44,−0.07] | −2.62 | 0.010 | 0.49 [0.00, 0.72] |
| | SVL | 0.56 [ 0.50, 0.62] | 0.88 [ 0.84, 0.91] | 17.52 | <0.001 | |
| Body fat | Brumation | 0.07 [-0.08, 0.22] | 0.10 [-0.11, 0.29] | 0.92 | 0.36 | 0.38 [0.00, 0.61] |
| | SVL | 1.43 [ 1.26, 1.60] | 0.88 [ 0.83, 0.91] | 17.08 | <0.001 | |
| Digestive tract | Brumation | 0.04 [-0.09, 0.20] | 0.06 [-0.15, 0.26] | 0.55 | 0.587 | 0.00 [0.00, 0.11] |
| | SVL | 0.99 [ 0.84, 1.13] | 0.83 [ 0.77, 0.87] | 14.03 | <0.001 | |
| Heart | Brumation | −0.07 [-0.20, 0.09] | −0.10 [-0.29, 0.11] | −0.93 | 0.355 | 0.13 [0.00, 0.33] |
| | SVL | 1.28 [ 1.14, 1.44] | 0.88 [ 0.83, 0.91] | 17.14 | <0.001 | |
| Lungs | Brumation | −0.11 [-0.27, 0.06] | −0.15 [-0.34, 0.06] | −1.40 | 0.165 | 0.23 [0.00, 0.46] |
| | SVL | 1.24 [ 1.08, 1.42] | 0.84 [ 0.78, 0.88] | 14.75 | <0.001 | |
| Liver | Brumation | 0.00 [-0.14, 0.16] | −0.01 [-0.21, 0.20] | −0.06 | 0.949 | 0.05 [0.00, 0.19] |
| | SVL | 1.13 [ 0.99, 1.29] | 0.85 [ 0.80, 0.89] | 15.38 | <0.001 | |
| Kidneys | Brumation | 0.03 [-0.08, 0.16] | 0.05 [-0.16, 0.25] | 0.48 | 0.635 | 0.00 [0.00, 0.11] |

*Appendix 1—table 9 Continued on next page*

*Appendix 1—table 9 Continued*

| Response | Predictors | β [95% CI] | Partial r [95% CI] | t | P | λ [95% CI] |
|---|---|---|---|---|---|---|
| | SVL | 1.15 [ 1.03, 1.27] | 0.90 [ 0.87, 0.93] | 19.88 | <0.001 | |
| Spleen | Brumation | –0.06 [-0.23, 0.13] | –0.07 [-0.26, 0.14] | –0.61 | 0.541 | 0.29 [0.00, 0.52] |
| | SVL | 1.32 [ 1.12, 1.53] | 0.82 [ 0.75, 0.86] | 13.32 | <0.001 | |
| Foreleg muscles | Brumation | –0.10 [-0.23, 0.06] | –0.16 [-0.35, 0.05] | –1.49 | 0.139 | 0.00 [0.00, 0.11] |
| | SVL | 1.41 [ 1.26, 1.55] | 0.91 [ 0.87, 0.93] | 20.34 | <0.001 | |
| Hindleg muscles | Brumation | –0.16 [-0.27,–0.04] | –0.30 [-0.47,–0.10] | –2.93 | 0.004 | 0.15 [0.00, 0.35] |
| | SVL | 1.27 [ 1.16, 1.40] | 0.92 [ 0.89, 0.94] | 21.76 | <0.001 | |
| Testes | Brumation | 0.19 [ 0.02, 0.35] | 0.25 [0.04, 0.42] | 2.38 | 0.019 | 0.55 [0.03, 0.76] |
| | SVL | 1.21 [ 1.01, 1.39] | 0.80 [0.72, 0.85] | 12.56 | <0.001 | |

**Appendix 1—table 10.** Effects of brumation on relative tissue size in males, using different temperature thresholds.

Phylogenetically controlled relationships ( λ = phylogenetic scaling parameter) between the brumation period and the relative male tissue mass in breeding condition (N = 116 species), with a 2 or 4°C buffer relative to *Appendix 1—table 7*. All associations are controlled for snout-vent length, confidence intervals are bootstrapped (N = 100 simulations).

| Response | Predictors | β [95% CI] | Partial r [95% CI] | t | P | λ [95% CI] |
|---|---|---|---|---|---|---|
| Brumation with 2 °C temperature buffer (df = 113) | | | | | | |
| Brain | Brumation | –0.08 [-0.14,–0.04] | –0.30 [-0.45,–0.12] | –3.29 | 0.001 | 0.35 [0.00, 0.61] |
| | SVL | 0.65 [ 0.59, 0.71] | 0.91 [ 0.87, 0.93] | 22.65 | <0.001 | |
| Body fat | Brumation | 0.17 [ 0.02, 0.29] | 0.21 [ 0.03, 0.37] | 2.29 | 0.024 | 0.44 [0.00, 0.69] |
| | SVL | 1.56 [ 1.38, 1.75] | 0.87 [ 0.83, 0.90] | 18.79 | <0.001 | |
| Digestive tract | Brumation | 0.03 [-0.09, 0.12] | 0.04 [-0.14, 0.22] | 0.43 | 0.667 | 0.00 [0.00, 0.11] |
| | SVL | 1.12 [ 1.01, 1.25] | 0.86 [ 0.81, 0.89] | 17.91 | <0.001 | |
| Heart | Brumation | –0.10 [-0.24, 0.01] | –0.14 [-0.31, 0.05] | –1.45 | 0.15 | 0.23 [0.00, 0.51] |
| | SVL | 1.35 [ 1.21, 1.51] | 0.87 [ 0.83, 0.90] | 18.84 | <0.001 | |
| Lungs | Brumation | –0.09 [-0.25, 0.03] | –0.11 [-0.29, 0.07] | –1.22 | 0.226 | 0.23 [0.00, 0.51] |
| | SVL | 1.32 [ 1.16, 1.50] | 0.83 [ 0.78, 0.87] | 16.13 | <0.001 | |
| Liver | Brumation | 0.01 [-0.11, 0.11] | 0.02 [-0.17, 0.20] | 0.17 | 0.867 | 0.00 [0.00, 0.11] |
| | SVL | 1.24 [ 1.13, 1.37] | 0.88 [ 0.84, 0.91] | 19.53 | <0.001 | |
| Kidneys | Brumation | –0.01 [-0.11, 0.08] | –0.01 [-0.19, 0.17] | –0.10 | 0.924 | 0.00 [0.00, 0.11] |
| | SVL | 1.24 [ 1.14, 1.36] | 0.90 [ 0.87, 0.92] | 22.33 | <0.001 | |
| Spleen | Brumation | –0.12 [-0.30, 0.02] | –0.13 [-0.30, 0.05] | –1.39 | 0.167 | 0.24 [0.00, 0.52] |
| | SVL | 1.34 [ 1.16, 1.55] | 0.80 [ 0.74, 0.85] | 14.38 | <0.001 | |
| Foreleg muscles | Brumation | –0.09 [-0.21, 0.01] | –0.13 [-0.30, 0.05] | –1.43 | 0.156 | 0.00 [0.00, 0.11] |
| | SVL | 1.53 [ 1.42, 1.66] | 0.91 [ 0.88, 0.93] | 23.80 | <0.001 | |
| Hindleg muscles | Brumation | –0.16 [-0.28,–0.08] | –0.28 [-0.43,–0.10] | –3.07 | 0.003 | 0.25 [0.00, 0.54] |
| | SVL | 1.35 [ 1.24, 1.48] | 0.91 [ 0.88, 0.93] | 23.46 | <0.001 | |
| Testes | Brumation | 0.25 [ 0.12, 0.39] | 0.31 [ 0.13, 0.46] | 3.46 | <0.001 | 0.75 [0.37, 0.89] |
| | SVL | 1.23 [ 1.02, 1.42] | 0.78 [ 0.71, 0.83] | 13.31 | <0.001 | |
| Brumation with 4 °C temperature buffer (df = 113) | | | | | | |

*Appendix 1—table 10 Continued on next page*

*Appendix 1—table 10 Continued*

| Response | Predictors | β [95% CI] | Partial r [95% CI] | t | P | λ [95% CI] |
|---|---|---|---|---|---|---|
| Brain | Brumation | –0.09 [-0.14,–0.05] | –0.33 [-0.47,–0.16] | –3.72 | <0.001 | 0.34 [0.00, 0.61] |
| | SVL | 0.65 [ 0.59, 0.71] | 0.91 [ 0.88, 0.93] | 23.01 | <0.001 | |
| Body fat | Brumation | 0.14 [ 0.00, 0.26] | 0.17 [-0.01, 0.34] | 1.86 | 0.065 | 0.44 [0.00, 0.69] |
| | SVL | 1.56 [ 1.38, 1.76] | 0.87 [ 0.82, 0.90] | 18.63 | <0.001 | |
| Digestive tract | Brumation | 0.03 [-0.08, 0.12] | 0.05 [-0.14, 0.22] | 0.48 | 0.63 | 0.00 [0.00, 0.11] |
| | SVL | 1.12 [ 1.02, 1.25] | 0.86 [ 0.81, 0.89] | 17.98 | <0.001 | |
| Heart | Brumation | –0.07 [-0.19, 0.04] | –0.10 [-0.27, 0.09] | –1.05 | 0.295 | 0.23 [0.00, 0.52] |
| | SVL | 1.35 [ 1.21, 1.50] | 0.87 [ 0.83, 0.90] | 18.67 | <0.001 | |
| Lungs | Brumation | –0.12 [-0.26, 0.00] | –0.15 [-0.32, 0.03] | –1.66 | 0.099 | 0.24 [0.00, 0.53] |
| | SVL | 1.33 [ 1.17, 1.51] | 0.84 [ 0.78, 0.87] | 16.25 | <0.001 | |
| Liver | Brumation | 0.01 [-0.11, 0.10] | 0.01 [-0.17, 0.19] | 0.10 | 0.92 | 0.00 [0.00, 0.11] |
| | SVL | 1.24 [ 1.14, 1.37] | 0.88 [ 0.84, 0.91] | 19.63 | <0.001 | |
| Kidneys | Brumation | 0.00 [-0.10, 0.08] | –0.01 [-0.19, 0.18] | –0.06 | 0.956 | 0.00 [0.00, 0.11] |
| | SVL | 1.24 [ 1.15, 1.35] | 0.90 [ 0.87, 0.93] | 22.41 | <0.001 | |
| Spleen | Brumation | –0.11 [-0.26, 0.03] | –0.12 [-0.29, 0.07] | –1.25 | 0.213 | 0.23 [0.00, 0.52] |
| | SVL | 1.34 [ 1.16, 1.54] | 0.80 [ 0.74, 0.85] | 14.34 | <0.001 | |
| Foreleg muscles | Brumation | –0.10 [-0.21,–0.01] | –0.15 [-0.32, 0.04] | –1.56 | 0.121 | 0.00 [0.00, 0.11] |
| | SVL | 1.53 [ 1.42, 1.66] | 0.91 [ 0.89, 0.93] | 23.95 | <0.001 | |
| Hindleg muscles | Brumation | –0.15 [-0.25,–0.07] | –0.26 [-0.42,–0.09] | –2.91 | 0.004 | 0.27 [0.00, 0.55] |
| | SVL | 1.35 [ 1.23, 1.48] | 0.91 [ 0.88, 0.93] | 23.24 | <0.001 | |
| Testes | Brumation | 0.22 [ 0.08, 0.34] | 0.28 [ 0.10, 0.43] | 3.10 | 0.002 | 0.76 [0.39, 0.90] |
| | SVL | 1.22 [ 1.01, 1.42] | 0.78 [ 0.70, 0.83] | 13.05 | <0.001 | |

**Appendix 1—table 11.** Species-specific environmental effects on the duration of the breeding season.

CV = coefficient of variation, P2T = duration of the dry season (in months), with a month defined as dry when its total precipitation is less than two times the mean temperature. All analyses based on df = 41. The regression slope β is presented with its bootstrapped confidence interval, λ is the phylogenetic scaling parameter.

| Predictor | β [95% CI] | r [95% CI] | t | P | λ [95% CI] |
|---|---|---|---|---|---|
| Brumation | –8.12 [-11.70,–5.01] | –0.57 [-0.72,–0.33] | –4.47 | <0.001 | 0.00 [0.00, 0.38] |
| Latitude | –4.86 [-8.95,–1.50] | –0.34 [-0.56,–0.05] | –2.33 | 0.025 | 0.00 [0.00, 0.39] |
| Longitude | 3.43 [-0.94, 8.44] | 0.24 [-0.06, 0.49] | 1.59 | 0.119 | 0.00 [0.00, 0.37] |
| Elevation | –5.56 [-9.84,–1.61] | –0.39 [-0.60,–0.10] | –2.73 | 0.009 | 0.00 [0.00, 0.38] |
| Mean temperature | 5.65 [1.78, 9.93] | 0.40 [0.11, 0.60] | 2.78 | 0.008 | 0.00 [0.00, 0.31] |
| Mean precipitation | 3.62 [-0.75, 7.62] | 0.26 [-0.05, 0.50] | 1.69 | 0.098 | 0.00 [0.00, 0.40] |
| CV temperature | –6.06 [-10.14,–2.06] | –0.43 [-0.62,–0.15] | –3.03 | 0.004 | 0.00 [0.00, 0.42] |
| CV precipitation | –1.44 [-5.70, 2.87] | –0.10 [-0.38, 0.20] | –0.65 | 0.518 | 0.00 [0.00, 0.42] |
| P2T | 2.73 [-1.28, 7.59] | 0.19 [-0.11, 0.45] | 1.25 | 0.217 | 0.00 [0.00, 0.40] |

**Appendix 1—table 12.** Species-specific environmental effects on the duration of the breeding season when accounting for brumation duration.

CV = coefficient of variation, P2T = duration of the dry season (in months), with a month defined as dry when its total precipitation is less than two times the mean temperature. All analyses based on df = 40. The regression slope β is presented with its bootstrapped confidence interval, $\lambda$ is the phylogenetic scaling parameter.

| Predictors | β [95% CI] | r [95% CI] | t | P | λ [95% CI] |
|---|---|---|---|---|---|
| Latitude | 2.01 [ –2.72, 6.60] | 0.12 [-0.19, 0.40] | 0.77 | 0.448 | 0.00 [0.00, 0.40] |
| Brumation | –9.56 [-14.76,–4.87] | –0.50 [-0.67,–0.23] | –3.64 | <0.001 | |
| Longitude | 1.35 [ –2.38, 4.50] | 0.11 [-0.20, 0.39] | 0.71 | 0.482 | 0.00 [0.00, 0.34] |
| Brumation | –7.76 [-10.87,–4.37] | –0.54 [-0.70,–0.29] | –4.09 | <0.001 | |
| Elevation | –2.12 [ –5.76, 1.96] | –0.16 [-0.43, 0.15] | –1.02 | 0.313 | 0.00 [0.00, 0.57] |
| Brumation | –7.09 [-12.59,–3.16] | –0.48 [-0.66,–0.20] | –3.41 | 0.001 | |
| Temperature | 2.45 [ –1.38, 5.77] | 0.19 [-0.12, 0.45] | 1.20 | 0.235 | 0.00 [0.00, 0.46] |
| Brumation | –7.00 [-11.08,–3.04] | –0.48 [-0.66,–0.21] | –3.44 | 0.001 | |
| Precipitation | 0.23 [ –3.31, 3.77] | 0.02 [-0.28, 0.31] | 0.12 | 0.909 | 0.00 [0.00, 0.27] |
| Brumation | –8.02 [-11.26,–4.34] | –0.53 [-0.69,–0.27] | –3.95 | <0.001 | |
| CV temperature | –2.41 [ –6.91, 1.41] | –0.18 [-0.44, 0.13] | –1.13 | 0.267 | 0.00 [0.00, 0.46] |
| Brumation | –6.83 [-10.85,–2.64] | –0.45 [-0.64,–0.17] | –3.19 | 0.003 | |
| CV precipitation | 0.64 [ –2.76, 3.95] | 0.05 [-0.25, 0.34] | 0.34 | 0.739 | 0.00 [0.00, 0.49] |
| Brumation | –8.28 [-11.95,–4.51] | –0.57 [-0.72,–0.32] | –4.36 | <0.001 | |
| P2T | 1.20 [ –2.12, 4.46] | 0.10 [-0.21, 0.38] | 0.64 | 0.524 | 0.00 [0.00, 0.44] |
| Brumation | –7.89 [-10.79,–3.66] | –0.56 [-0.71,–0.31] | –4.23 | <0.001 | |

**Appendix 1—table 13.** Comparison of candidate models in a phylogenetic path analysis linking environmental variation to brumation duration and the breeding patterns (breeding season and aggregations).

All candidate models are ranked according to their C-statistic Information Criterion (CICc). Those with ΔCICc < 2 are highlighted in bold and were used to calculate the average model, which is depicted in *Appendix 1—figure 4* along with all hypotheses.

| Model | k | q | C | P | CICc | ΔCICc | w_i |
|---|---|---|---|---|---|---|---|
| m5 | 3 | 7 | 3.32 | 0.77 | 20.52 | 0.00 | 1.00 |
| m4 | 2 | 8 | 1.72 | 0.79 | 21.95 | 1.43 | 0.49 |
| m1 | 1 | 9 | 0.29 | 0.87 | 23.74 | 3.22 | 0.20 |
| m3 | 2 | 8 | 15.06 | 0.00 | 35.29 | 14.77 | 0.00 |
| m6 | 3 | 7 | 22.03 | 0.00 | 39.23 | 18.71 | 0.00 |
| m2 | 3 | 7 | 35.60 | 0.00 | 52.80 | 32.28 | 0.00 |
| m8 | 3 | 7 | 60.09 | 0.00 | 77.29 | 56.77 | 0.00 |
| m7 | 2 | 8 | 83.01 | 0.00 | 103.25 | 82.73 | 0.00 |

*k* = number of independence claims; *q* = number of parameters; *C* = Fisher's *C* statistics; CICc = C-statistic Information Criterion; ΔCICc = difference in CICc from the best-fitting model; *w_i* = CICc weight.

**Appendix 1—table 14.** Correlation matrix between the compositional data of the four focal traits and the rest of the body.

The upper triangle lists the correlation coefficients with 95% confidence intervals (controlling for phylogeny), the lower triangle the corresponding P-values before (upper) and after (lower) correction for multiple testing based on the false discovery rate. Statistically significant correlations

are highlighted in bold.

| | Brain | Body fat | Testes | Hindlimb muscles | Rest |
|---|---|---|---|---|---|
| Brain | | –0.79 [-0.83,–0.71] | –0.36 [-0.50,–0.19] | –0.08 [-0.26, 0.10] | 0.30 [0.12, 0.44] |
| Body fat | <0.001 < 0.001 | | –0.06 [-0.23, 0.12] | –0.03 [-0.20, 0.15] | –0.17 [-0.34, 0.01] |
| Testes | <0.001 < 0.001 | 0.531 0.590 | | –0.54 [-0.64,–0.40] | –0.57 [-0.67,–0.44] |
| Hindlimb muscles | 0.370 0.527 | 0.776 0.776 | <0.001 < 0.001 | | 0.07 [-0.11, 0.24] |
| Rest | 0.001 0.002 | 0.060 0.101 | <0.001 < 0.001 | 0.466 0.582 | |

**Appendix 1—table 15.** Results of the phylogenetic principal component analysis.
Shown are the eigenvalues and percent of the total variance explained by the four principal components, as well as the loadings of the original variables. Loadings >0.25 are highlighted in bold. Phylogenetic scaling parameter $\lambda$ = 0.75.

| | PC1 | PC2 | PC3 | PC4 |
|---|---|---|---|---|
| Eigenvalues | 3.38 | 0.32 | 0.21 | 0.09 |
| % variance | 84.4 | 8.0 | 5.2 | 2.3 |
| | | | | |
| Loadings | | | | |
| Brain | –0.91 | 0.31 | 0.26 | –0.12 |
| Body fat | –0.93 | –0.07 | –0.31 | –0.16 |
| Testes | –0.88 | –0.44 | 0.18 | 0.04 |
| Hindlimb muscles | –0.95 | 0.18 | –0.11 | 0.23 |

**Appendix 1—table 16.** Phylogenetic path analysis linking brumation to relative trait investments. All candidate models are ranked according to their CICc. Those with ΔCICc < 2 are highlighted in bold and were used to calculate the average model (see *Figure 4* in main text). For candidate models and averaged path coefficients see *Appendix 1—figures 6 and 7*.

| Model | k | q | C | P | CICc | ΔCICc | $w_i$ |
|---|---|---|---|---|---|---|---|
| m11 | 4 | 17 | 8.28 | 0.41 | 48.50 | 0.00 | 0.25 |
| m25 | 1 | 20 | 0.99 | 0.61 | 49.80 | 1.31 | 0.13 |
| m23 | 1 | 20 | 1.28 | 0.53 | 50.10 | 1.59 | 0.11 |
| m24 | 1 | 20 | 1.28 | 0.53 | 50.10 | 1.59 | 0.11 |
| m3 | 3 | 18 | 7.44 | 0.28 | 50.50 | 1.96 | 0.09 |
| m12 | 3 | 18 | 7.53 | 0.28 | 50.60 | 2.05 | 0.09 |
| m9 | 5 | 16 | 13.48 | 0.20 | 51.00 | 2.45 | 0.07 |
| m21 | 2 | 19 | 6.52 | 0.16 | 52.40 | 3.91 | 0.03 |
| m4 | 2 | 19 | 6.68 | 0.15 | 52.60 | 4.08 | 0.03 |
| m1 | 4 | 17 | 12.64 | 0.13 | 52.90 | 4.35 | 0.03 |
| m10 | 4 | 17 | 12.72 | 0.12 | 53.00 | 4.44 | 0.03 |
| m22 | 2 | 19 | 7.54 | 0.11 | 53.50 | 4.93 | 0.02 |
| m2 | 3 | 18 | 11.88 | 0.06 | 54.90 | 6.41 | 0.01 |
| m26 | 3 | 18 | 14.45 | 0.03 | 57.50 | 8.98 | 0.00 |
| m7 | 4 | 17 | 23.32 | 0.00 | 63.60 | 15.04 | 0.00 |
| m19 | 4 | 17 | 23.95 | 0.00 | 64.20 | 15.66 | 0.00 |

*Appendix 1—table 16 Continued on next page*

*Appendix 1—table 16 Continued*

| Model | k | q | C | P | CICc | ΔCICc | wᵢ |
|-------|---|---|-----|------|-------|-------|------|
| m8 | 3 | 18 | 22.57 | 0.00 | 65.60 | 17.09 | 0.00 |
| m5 | 5 | 16 | 28.52 | 0.00 | 66.00 | 17.49 | 0.00 |
| m14 | 3 | 18 | 23.59 | 0.00 | 66.60 | 18.11 | 0.00 |
| m13 | 4 | 17 | 26.53 | 0.00 | 66.80 | 18.25 | 0.00 |
| m20 | 3 | 18 | 24.03 | 0.00 | 67.10 | 18.56 | 0.00 |
| m6 | 4 | 17 | 27.76 | 0.00 | 68.00 | 19.48 | 0.00 |
| m16 | 5 | 16 | 36.26 | 0.00 | 73.80 | 25.23 | 0.00 |
| m27 | 5 | 16 | 37.41 | 0.00 | 74.90 | 26.38 | 0.00 |
| m15 | 3 | 18 | 34.57 | 0.00 | 77.60 | 29.10 | 0.00 |
| m17 | 4 | 17 | 37.68 | 0.00 | 77.90 | 29.39 | 0.00 |
| m28 | 6 | 15 | 50.57 | 0.00 | 85.40 | 36.85 | 0.00 |
| m18 | 4 | 17 | 59.08 | 0.00 | 99.30 | 50.80 | 0.00 |

k = number of independence claims; q = number of parameters; C = Fisher's C statistics; CICc = C-statistic Information Criterion; ΔCICc = difference in CICc from the best-fitting model; wᵢ = CICc weight.

**Appendix 1—table 17.** Effect of the predominant habitat of each species on the relative amount of body fat in the context of the brain−fat trade-off hypothesis.

Output of PGLS models examining the effect of the predominant species-specific habitat (and so mode of locomotion) on the relative amount of body fat, accounting for brain mass and brumation duration as other sources of variation. Three different models were conducted: (1) using raw species means with log body fat as the response variable and controlling for body size by including log snout-vent length as a covariate; (2) using the compositional data (centred log ratio of both brain mass as the response and brain mass); and (3) using PC3 of the principal component analysis, which was primarily loaded by brain mass (+0.26) and body fat (−0.31; *Figure 3*). Statistically significant effects are highlighted in bold, and the different habitat types are compared against aquatic species as the reference in the intercept.

| Predictors | β [95% CI] | r [95% CI] | t | P |
|------------|------------|------------|-----|-----|
| Raw species means (df = 109, $\lambda$ =0.17 [0.00, 0.51]) | | | | |
| (Intercept) | −12.97 [−17.49,−8.41] | −0.44 [-0.57,−0.28] | −5.12 | <0.001 |
| Log Brain mass | 0.50 [0.12, 0.94] | 0.18 [0.00, 0.35] | 1.94 | 0.055 |
| Habitat [arboreal] | −0.97 [−1.49,−0.38] | −0.3 [-0.45,−0.12] | −3.31 | 0.001 |
| Habitat [semiaquatic] | −0.02 [-0.31, 0.35] | −0.01 [-0.19, 0.18] | −0.09 | 0.925 |
| Habitat [terrestrial] | −0.3 [-0.83, 0.35] | −0.11 [-0.28, 0.08] | −1.12 | 0.264 |
| Brumation | 0.22 [0.09, 0.37] | 0.27 [0.09, 0.42] | 2.90 | 0.005 |
| Log SVL | 2.94 [2.13, 3.76] | 0.52 [0.37, 0.63] | 6.34 | <0.001 |
| Compositional data (df = 110, $\lambda$ =0.01 [0.00, 0.03]) | | | | |
| (Intercept) | −1.94 [−2.14,−1.69] | −0.86 [-0.89,−0.82] | −17.96 | <0.001 |
| Brain mass | −0.61 [-0.74,−0.52] | −0.68 [-0.76,−0.58] | −9.81 | <0.001 |
| Habitat [arboreal] | −0.28 [-0.51,−0.06] | −0.22 [-0.38,−0.04] | −2.39 | 0.018 |
| Habitat [semiaquatic] | −0.10 [-0.31, 0.06] | −0.10 [-0.28, 0.09] | −1.07 | 0.286 |
| Habitat [terrestrial] | 0.01 [-0.19, 0.20] | 0.01 [-0.17, 0.19] | 0.12 | 0.901 |
| Brumation | 0.08 [0.01, 0.17] | 0.19 [0.00, 0.35] | 1.98 | 0.050 |

*Appendix 1—table 17 Continued on next page*

*Appendix 1—table 17 Continued*

| Predictors | β [95% CI] | r [95% CI] | t | P |
|---|---|---|---|---|
| PC3 (df = 111, $\lambda$ =0.20 [0.00, 0.59]) | | | | |
| (Intercept) | −0.03 [-0.26, 0.16] | −0.06 [-0.24, 0.12] | −0.27 | 0.785 |
| Habitat [arboreal] | 0.31 [0.12, 0.52] | 0.26 [0.08, 0.42] | 2.83 | 0.006 |
| Habitat [semiaquatic] | −0.01 [-0.15, 0.13] | −0.01 [-0.19, 0.17] | −0.10 | 0.920 |
| Habitat [terrestrial] | 0.10 [-0.06, 0.30] | 0.09 [-0.09, 0.27] | 0.99 | 0.326 |
| Brumation | −0.02 [-0.08, 0.02] | −0.08 [-0.26, 0.10] | −0.88 | 0.381 |

