## [Editor Report · eLife assessment]

In this **important** paper, the authors report a link between brumation (or "hibernation") and tissue size in frogs, summarizing **convincing** evidence that extended brumation is associated with smaller brain size and increased investment in reproduction-related tissues. The research is of broad interest to ecologists, evolutionary biologists, and those interested in global change biology, as the dataset involves significant field work and advanced statistical analyses for insights into how expensive tissues in these ectothermic animals respond to environmental seasonality.

---

## [Referee Report · Joint Public Review]

The authors have greatly improved the manuscript after detailed revisions. I would like to discuss with the authors on how to make their findings more general across taxonomic groups. For example, whether it is possible for authors to conduct a more comprehensive analyses by including amphibians, birds, and mammals together to test the general role of the relationship between brain evolution and environmental resources, and what ecological factors determine the observed brain size variations among taxa except for their biological differences such as energetic demands. It is especially for population-level analyses when related data is available in the future, which may provide very helpful insights into the brain size biogeographic patterns and their determinants across taxa.

---

## [Author Response]

The following is the authors’ response to the original reviews.

**eLife assessment**
In this important paper, the authors report a link between brumation and tissue size in frogs, summarizing convincing evidence that extended brumation is associated with smaller brain size and increased investment in reproduction-related tissues. The research will be of broad interest to ecologists, evolutionary biologists, and those interested in global change biology. While the dataset involves significant field work and advanced statistical analyses, the manuscript would benefit from more explanation of the models, including why frogs are a good model in which to address these questions, and from general improvement in the structure and conciseness.

We highly appreciate your positive assessment and that you considered our paper important and convincing.

**Reviewer #1 (Public Review):**
The authors have conducted lots of field work, lab work and statistical analysis to explore the effect of brumation on individual tissue investments, the evolutionary links between the relative costly tissue sizes, and the complex non-dependent processes of brain and reproductive evolution in anuran. The topic fits well within the scope of the journal and the manuscript is generally written well. The different parameters used in the present study will attract a board readership across ecology, zoology, evolution biology, and global change biology.

Thank you for your positive and supporting feedback.

**Reviewer #2 (Public Review):**
The authors set out to show how hibernation is linked to brain size in frogs. If there were broader aims it is hard to decipher them. The authors present an extremely impressive dataset and a thorough set of cutting-edge analyses. However not all details are well explained. The main result about hibernation and brain size is fairly convincing, but it is hard to think of broader implications for this study. Overall, the manuscript is very confusing and hard to follow.

Thank you for your compliments on our paper. As for your concerns, we have greatly revised our paper and, as we hope, improved its clarity. We have also added a few sentences to the conclusions to draw attention to potentially broader implications. Specifically, we describe how the focal traits of our study may all be affected by climate change. Differential constraints in necessary investments could be one of several reasons for the varying resilience to climate change between species in the same habitat.

**Reviewer #1 (Recommendations For The Authors):**
There are no issues on the availability of data and code.

Thank you.

Line 15: in the author contribution section, it seems that C.L.M. and J.P.Y are not in the author list.

These two authors are not part of this study. This was a mistake.

Line 24: I don't think it is vital or logical to address or compare too much on birds or mammals, which are not the focused taxa of the present study. Instead, it is better to clarify the reason why frogs and toads are ideal model taxon to this study.

The reason for comparisons with birds and mammals was that all hypotheses related to the various trade-offs tested here had been developed in these taxa. One of the points of our paper was that these needed validation beyond the two taxa, in addition to being tested against one another (each prediction had been developed in a specific group and typically in isolation of all other hypotheses).

Line 25-26: as the authors are shooting for eLife, as a general journal, it is not essential to provide the detailed methods in the abstract. But I think the authors need to strengthen the novelty of the work in the field here.

The strength of our study was that all traits were measured directly in our species, including estimates of hibernation duration. Prior studies used various proxies, categorial classification or datasets assembled from multiple sources. To us, this seemed like a sufficiently important advance in the field to mention it, but considering the reviewer’s comment, we have now removed it.

Line 28: "protracted brumation reduces brain size and instead promotes reproductive investments", as a correlative study, it is much more precise to change this sentence to a similar description as "protracted brumation is negatively correlated with brain size but is positively correlated with reproductive investments" here and related statements throughout the whole text.

We agree that, strictly speaking, a path analysis can only point toward possible causality and not provide hard evidence as experimental manipulation might. The wording may have been a bit too strong here in our attempt to minimize wordiness and because all our analyses combined very strongly pointed in this direction. However, we have now changed this as suggested even though it now reads almost as if we had done no more than conducting a simple correlation. We have further paid attention to the wording of our interpretations throughout the paper.

Line 32-33: it needs a bigger ending linking your main findings with the implication in understanding species response to the sustained environment change.

We have reworded the ending of the abstract to: “Our results provide novel insights into resource allocation strategies and possible constraints in trait diversification, which may have important implications for the adaptability of species under sustained environmental change.”

Line 63-68: this sentence is too long to understand and please simplify it.

We have split the sentence into two sentences.

Line 125-130: it is known that there are various frog reproductive modes (Crump et al. 2015) such as trade-offs between clutch size and egg size, different number of breeding during one year, etc. These different reproductive forms may also influence the brain size evolution with food availability and seasonal variations. Please clarify it.

Yes, anurans do have varying reproductive modes, but to us, there is no a priori reason to assume that such variation would have a direct effect on brain evolution. Rather, in our opinion, different reproductive modes would have indirect effects by affecting the environment in which reproduction occurs. For example, larvae developing under different environmental conditions (substrate, larval density, egg provisioning etc.) might affect developmental trajectories that could influence how resources are available and allocated to different organs, including the brain. Alternatively, reproductive modes could influence the choice of environment for reproduction, thereby possibly affecting mating strategies and ultimately trait investments associated with these strategies. Given we were asked to shorten our paper, we believe that ‘environmental effects’ remains broad enough to encompass such variation, thereby not necessitating disentangling the different, and likely primarily indirect, ways that reproductive modes could be linked to brain evolution. However, if the reviewer would find it important to go into such detail in the paper, we will be happy to do so.

Line 186-187: it is necessary to mention here that the authors also conducted sensitivity analyses to apply 2{degree sign}C or 4{degree sign}C below their experimentally derived as thresholds to test the robustness of the results to data uncertainty.

We have added “(details on methodology and various sensitivity analyses for validation in Material and Methods)” to indicate the different types of sensitivity analyses, which included more than simply 2 or 4°C difference.

Line 188: please change "In phylogenetic regressions" to "after controlling for phylogenetic autocorrelation/pseudo-replication" or similar sentence here.

Our focus here was the phylogenetically informed GLS model rather than phylogenetic control itself. In the latter case, it would still not be clear what type of model was conducted with such phylogenetic control. To avoid any shorthand, we have reworded for more precision: “We employed phylogenetic generalized least-squares (PGLS) models, …”

Line 177-287: please provide the exact variance explained by different predictor variables in brumation duration, individual tissue investments, and brain evolution. I also suggest that the authors need consider conducting multi-model inference-based model averaging analysis to test the relative importance of different variables. In addition, the present analyses did not include the interaction terms among variables, which may be more important than the effect of each individual factor.

There may be some misunderstanding as these models represent separate analyses for each predictor as indicated by the associated λ values (never more than one value per model). We conducted separate models to determine which variables might even play a role in explaining variation in the corresponding response variables. Based on relevant predictors, we then conducted path analyses rather than general multi-predictor analyses. The relative effect sizes are represented by the correlation coefficients (r values) in the tables.

**Reviewer #2 (Recommendations For The Authors):**
Why exactly are the pairwise comparisons positively correlated (fig. S5) and then negatively correlated (fig. 3). What is actually driving this difference? For the phylogenetic path analyses 26 candidate models are chosen without explanation. What theory or hypotheses are these based on?

We assume the reviewer is referring to the brain-body fat association. The two ‘pairwise’ analyses they mention were not the same. The correlation in Fig. S5 was a standard (albeit phylogenetically informed) partial correlation between the two focal tissues, controlling for SVL. By contrast, as described when introducing the analyses, negative associations were derived when additionally controlling for testes and hindlimb muscles, all of which deviated from isometry against body size. Here, the total mass of the four main tissues was divided by their proportional contribution to that mass in each species, then standardized for comparison across species. Since the total mass of these four tissues scaled directly with body size, larger-bodied species did not invest a proportion of their body to these tissues than smaller-bodied species, thus essentially rendering body size irrelevant for this analysis. However, the relative representation of the four traits changed between species such that more resources devoted to body fat was associated with a smaller brain, hence a negative relationship. Similarly, the multivariate analysis as well as the PCA also suggested similar trends when all four tissues were considered rather than purely pairwise comparisons.

Regarding the second comment: We indeed used 28 pre-defined predictions for our larger path analysis.

The authors haven't really provided much additional context either, and the discussion is almost entirely a rehash of the results section. I can't see the analysis code but this may be of use to people performing similar analyses.

It is true that the traits and core message of the Discussion relate directly to our results, but we believe that our Discussion provides the essential biological context to our findings and to how they are connected. We tried not to go on tangents or too much speculation as the many results provided enough material to discuss, with several different ways that we expanded the prior state-of-the-art in the field. However, we have now expanded the concluding paragraph to place our findings in the context of climate change, given that this could affect anurans and the different traits examined in many ways that are directly related to the current study. Yet, we decided to keep this short because such extrapolation of our findings

We indeed held off making the code available to the public in case dramatic changes to the paper were requested by the reviewers. However, it will be published.

Additional recommendations from the Reviewing Editor:One of the reviewers and I found the text a little difficult to follow. I suggest simplifying the paper by being more concise. For example, the introduction could be shortened into a 3-4 paragraphs of relevant text without overwhelming the reader. One of the reviewers wanted a better explanation of statistical models and I agree. The discussion could benefit from some structure - consider adding subheadings that would guide the reader as to the topic. Finally, the figures are difficult to see and should be made larger. For example, the graphs in Figure 1c could be on a panel below A and B so that readers can interpret the graph. In Figure 3 - the legend is far too small - please put above or below the graphs. In summary - I hope you consider a major re-write that would strengthen the accessibility of your paper to a broad audience.

We have substantially shortened the paper despite adding further details on models and a broader context to the Discussion. We also condensed the Introduction to about two thirds of the original word count. However, we did not think that shortening it even further or splitting it into 3-4 paragraphs would improve readability. We still considered it important to introduce with sufficient context all major hypotheses that were tested against one another, provide at least some information on what was or was not known about the evolution of the focal traits and their links to one another or the environmental variables. We also found it important to touch on the differences between our study organisms and those typically studied in the context of hibernation or brain evolution, as this could affect the predictions. Given the number of hypotheses and traits, cutting the number of paragraphs would have meant merging some of them into very long ones, which we did not consider helpful.

We further added short subheadings to the Discussion and adjusted the figures as requested.